# A stabilization rule for metal carbido cluster bearing μ₃-carbido single-atom-ligand encapsulated in carbon cage

Runnan Guan [1,5], Jing Huang[2,3,5], Jinpeng Xin[1], Muqing Chen [1], Pingwu Du [1], Qunxiang Li [2] ✉, Yuan-Zhi Tan [4] ✉, Shangfeng Yang [1] ✉ & Su-Yuan Xie [4]

Metal carbido complexes bearing single-carbon-atom ligand such as nitrogenase provide ideal models of adsorbed carbon atoms in heterogeneous catalysis. Trimetallic μ₃-carbido clusterfullerenes found recently represent the simplest metal carbido complexes with the ligands being only carbon atoms, but only few are crystallographically characterized, and its formation prerequisite is unclear. Herein, we synthesize and isolate three vanadium-based μ₃-CCFs featuring V = C double bonds and high valence state of V (+4), including $VSc_2C@I_h(7)\text{-}C_{80}$, $VSc_2C@D_{5h}(6)\text{-}C_{80}$ and $VSc_2C@D_{3h}(5)\text{-}C_{78}$. Based on a systematic theoretical study of all reported μ₃-carbido clusterfullerenes, we further propose a supplemental Octet Rule, i.e., an eight-electron configuration of the μ₃-carbido ligand is needed for stabilization of metal carbido clusters within μ₃-carbido clusterfullerenes. Distinct from the classic Effective Atomic Number rule based on valence electron count of metal proposed in the 1920s, this rule counts the valence electrons of the single-carbon-atom ligand, and offers a general rule governing the stabilities of μ₃-carbido clusterfullerenes.

Organometallic complexes play a crucial role in catalysis, energy and medicine nowadays. Stabilities of organometallic complexes have been commonly determined by Effective Atomic Number (EAN) rule (i.e., 18-electron rule) proposed in the 1920s, that the effective atomic number of the central metal atom surrounded by ligands is numerically equal to the atomic number of the noble-gas element found in the same period as the metal[1]. EAN rule is based on valence electron count of the central metal atom instead of the non-metal ligand, and is applicable for a majority of organometallic complexes. In particular, metal carbido complexes bearing single-carbon-atom ligand such as the active site of nitrogenase ($Fe_7MoS_9C$) provide ideal models of adsorbed carbon atoms in heterogeneous catalysis, and have been attracting enormous interests during the past few decades[2–10]. Unlike the traditional multinuclear organometallic complexes, for metal carbido complexes the single-carbon-atom ligand becomes the center and bonds with 1 to 6 metals, thus the EAN rule is inapplicable due to the complicated coordination nature of the ligands especially the central single-carbon-atom ligand[2–10]. Hence, it is desirable to establish a new rule governing the stabilities of metal carbido complexes. For binuclear metal carbido complexes containing a carbido bridge such as $L_nM = C = ML_n$ and $L_nM \equiv C\text{-}M'L_n$, the central μ₂-carbido ligand adopts an eight-electron configuration[4,5]. However, upon increasing

[1]Key Laboratory of Precision and Intelligent Chemistry, Collaborative Innovation Center of Chemistry for Energy Materials (iChEM), Department of Materials Science and Engineering, University of Science and Technology of China, Hefei 230026, China. [2]Hefei National Laboratory for Physical Sciences at Microscale, Department of Chemical Physics, Synergetic Innovation Center of Quantum Information & Quantum Physics, University of Science and Technology of China, Hefei 230026, China. [3]School of Materials and Chemical Engineering, Anhui Jianzhu University, Hefei 230601, China. [4]State Key Lab for Physical Chemistry of Solid Surfaces, Collaborative Innovation Center of Chemistry for Energy Materials (iChEM), Department of Chemistry, College of Chemistry and Chemical Engineering, Xiamen University, Xiamen 361005, China. [5]These authors contributed equally: Runnan Guan, Jing Huang. ✉ e-mail: liqun@ustc.edu.cn; yuanzhi_tan@xmu.edu.cn; sfyang@ustc.edu.cn

the number of metals coordinated with the central single-carbon-atom ligand to three, the central $\mu_3$-carbido ligand does not always follow the eight-electron configuration[6–8]. Furthermore, in the metal carbido complexes bearing $\mu_5$- and $\mu_6$-carbido ligands, the central carbon atom is regarded as a hypervalency carbon due to the formation of more than four metal-carbon bonds[9,10]. Therefore, due to the diversity of the coordination numbers of the central single-carbon-atom ligand, it is difficult to establish a general rule for the conventional metal carbido complexes.

As the simplest metal carbido complexes with the ligands being only carbon atoms, trimetallic $\mu_3$-carbido clusterfullerenes ($\mu_3$-CCFs) featuring confinement of a single-carbon-atom ligand within carbon cage was found in 2014 (ref. 11). Due to electron transfer from the encapsulated metal carbido cluster to the outer carbon cage, $\mu_3$-CCFs exhibit intriguing electronic properties and promising applications in spintronics and high-density storage devices inaccessible by the conventional metal carbido complexes[11–19]. $TiLu_2C@I_h(7)\text{-}C_{80}$ is the first $\mu_3$-CCF isolated in 2014, in which a Ti=C double bond was identified by single-crystal X-ray diffraction[11]. Later on, a few Ti-based $\mu_3$-CCFs were isolated, including $TiM_2C@I_h(7)\text{-}C_{80}$ (M = Sc, Y, Nd, Gd, Tb, Dy, Er)[11–16], $TiM_2C@D_{5h}(6)\text{-}C_{80}$ (M = Sc, Dy)[12,13], and $TiSc_2C@C_{78}$ (ref. 13), among which only $TiSc_2C@I_h(7)\text{-}C_{80}$, $TiDy_2C@I_h(7)\text{-}C_{80}$ and $TiTb_2C@I_h(7)\text{-}C_{80}$ were crystallographically determined. More recently, another non-rare earth (non-RE) metal, the actinide metal uranium (U), was also reported to form $\mu_3$-CCF $USc_2C@I_h(7)\text{-}C_{80}$, in which the U metal exhibits a formal valence state of +4 and bonds with the central $\mu_3$-carbido ligand via a U = C double bond as well[17,18]. Different to the well-known trimetallic nitride clusterfullerenes (NCFs) $M_3N@C_{2n}$ bearing primarily RE metals with +3 valence states ($M^{3+}$) and M-N single bonds[20,21], in $\mu_3$-CCFs the valence state of the non-RE metal changes to +4 as the result of formation of M = C (M = Ti, U) double bond, while the RE metals keep the +3 valence states and M-N single bonds[11–19]. Hence, $\mu_3$-CCFs offer a unique platform stabilizing $\mu_3$-carbido ligand which bonds with metals via multiple bonds. Although a number of $\mu_3$-CCFs have been isolated, only few were crystallographically characterized, and the reported non-RE metals within $\mu_3$-CCFs are quite limited to Ti and U. This limitation is because the formation prerequisite of $\mu_3$-CCF is unclear. Therefore, it is highly desired to explore new $\mu_3$-CCFs based on other non-RE metals and to establish a general rule elucidating stabilization of the $\mu_3$-carbido ligand within it.

Herein, we synthesized and isolated three vanadium(V)-based $\mu_3$-CCFs, including $VSc_2C@I_h(7)\text{-}C_{80}$, $VSc_2C@D_{5h}(6)\text{-}C_{80}$ and $VSc_2C@D_{3h}(5)\text{-}C_{78}$, among them the latter two represent the first crystallographically determined non-$I_h$-symmetry $\mu_3$-CCFs. The common feature of their molecular structures is the existence of V = C double bond along with Sc-C single bonds. Their electronic structures were investigated by density functional theory (DFT) calculations, unraveling high valence state of V (+4). Combining all reported sixteen $\mu_3$-CCFs, we further carried out a systematic DFT study on their stabilities, and proposed a supplemental Octet Rule based on valence electron count of the central ligand to account for stabilization of the metal carbido cluster within $\mu_3$-CCF. This rule is also applicable for NCF $VSc_2N@C_{80}$ and the conventional binuclear metal carbido complexes containing a carbido bridge such as $L_nM = C=ML_n$ and $L_nM \equiv C\text{-}M'L_n$, thus offers a general rule determining the stabilities of $\mu_3$-CCFs and guides the exploration of $\mu_3$-CCFs or even other metal carbido complexes.

## Results

### Syntheses and Isolation of $VSc_2C@C_{80}$ (I, II) and $VSc_2C@C_{78}$

V-based $\mu_3$-CCFs, including two isomers of $VSc_2C@C_{80}$ (labeled as I, II) and $VSc_2C@C_{78}$ were synthesized by Krätschmer-Huffman direct current (DC) arc discharge method[19]. Graphite rods packed with a mixture of $Sc_2O_3$, VC and graphite powder with a molar ratio of 0.5:1:15 were vaporized in the arcing chamber under a 200 mbar helium atmosphere. The obtained soot was then extracted with carbon disulfide ($CS_2$), followed by four-step high-performance liquid chromatography (HPLC) isolations supplemented by laser desorption time-of-flight mass spectroscopic (LD-TOF MS) analysis. In the first step, fractions **A** and **B** both contain the same MS signal peak at M/Z = 1113 (Supplementary Fig. 1 and Table 1), which is assigned to $VSc_2C@C_{80}$. Since the retention times of fractions **A** and **B** are quite different, the two $VSc_2C@C_{80}$ molecules detected in these two fractions are isomers with different cage isomeric structures. The first isomer $VSc_2C@C_{80}$ (I) has been isolated from fraction **A** and identified as $VSc_2C@I_h(7)\text{-}C_{80}$ very recently (Supplementary Fig. 2)[19], therefore the second isomer isolated from fractions **B** after four-step HPLC separation is labeled as $VSc_2C@C_{80}$ (II) (Supplementary Fig. 3). Besides, another V-$\mu_3$-CCF with a MS signal peak at M/Z = 1089 is also isolated from fraction **C** (Supplementary Fig. 4), which is assigned to $VSc_2C@C_{78}$.

The high purities of the isolated $VSc_2C@C_{80}$ (II) and $VSc_2C@C_{78}$ are verified by the single peaks observed by HPLC (Fig. 1a) and single mass peaks in their LD-TOF MS spectra (Fig. 1b). Furthermore, the isotopic distributions of $VSc_2C@C_{80}$ (II) and $VSc_2C@C_{78}$ agree well with the calculated ones, confirming their proposed chemical formulae. Interestingly, the analogous V-based $\mu_3$-CCF $V_2ScC@C_{80}$ and

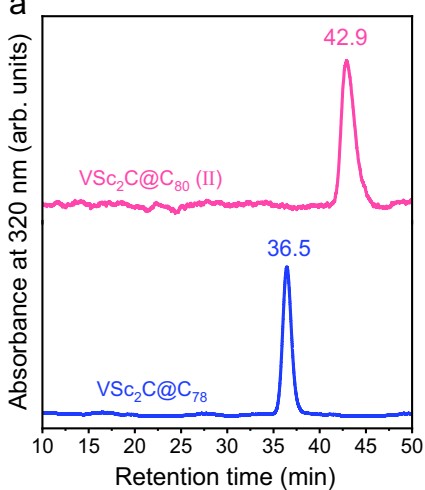
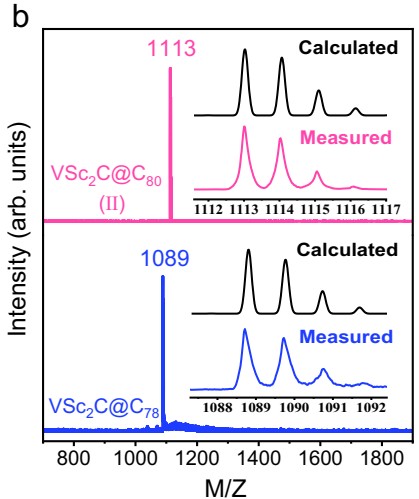

**Fig. 1 | The purity examination of $VSc_2C@C_{80}$ (II) and $VSc_2C@C_{78}$. a** HPLC chromatograms and **b** LD-TOF mass spectra of $VSc_2C@C_{80}$ (II) and $VSc_2C@C_{78}$. (column: Buckyprep, eluent: toluene, flow rate: 5 mL.min$^{-1}$, injection volume: 5 mL, temperature: 40 °C).

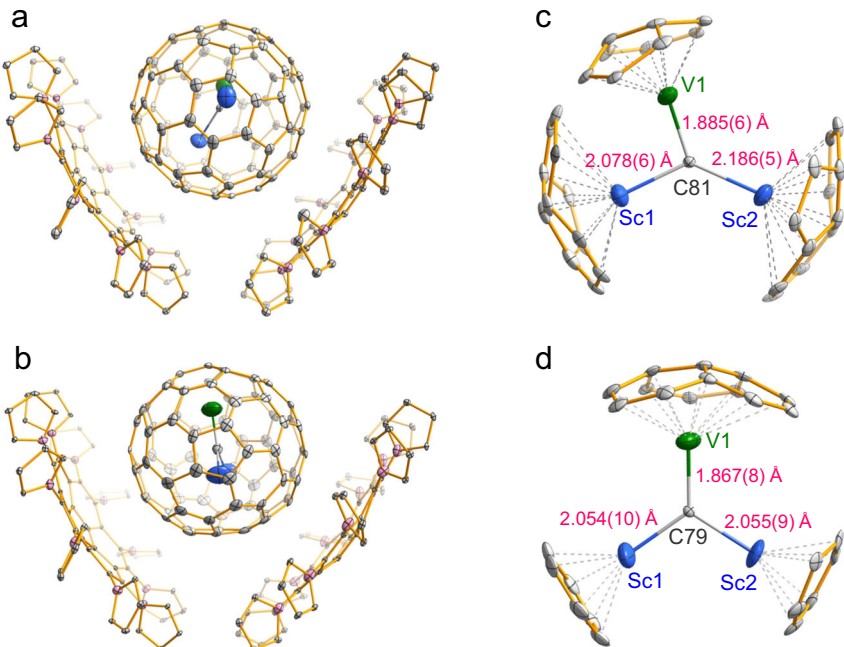

**Fig. 2 | X-ray structures of VSc₂C@_D₅ₕ_(6)-C₈₀ and VSc₂C@_D₃ₕ_(5)-C₇₈.** Drawings of the crystallographically determined structures of VSc₂C@_D₅ₕ_(6)-C₈₀·2(DPC) (**a**) and VSc₂C@_D₃ₕ_(5)-C₇₈·2(DPC) (**b**). The positions of the major cluster sites with respect to the nearest carbon atoms of cage within VSc₂C@_D₅ₕ_(6)-C₈₀ (**c**) and

VSc₂C@_D₃ₕ_(5)-C₇₈ (**d**). Only one orientation of the fullerene cage together with the major site of VSc₂C cluster is given for clarity. Solvent molecules, the minor cage and minor metal positions are omitted for clarity. Gray: C; Blue: Sc; Green: V; Pink: N.

Sc-only μ₃-CCF Sc₃C@C₈₀ are not detected, and this phenomenon is distinctly different from the case of the reported V-based NCFs for which both VSc₂N@C₈₀ and V₂ScN@C₈₀ were synthesized along with Sc₃N@C₈₀[22,23]. The difference between V-based μ₃-CCFs and NCFs suggests the unique formation prerequisite of μ₃-CCF as discussed in details below.

### X-ray crystallographic structures of VSc₂C@_D₅ₕ_(6)-C₈₀ and VSc₂C@_D₃ₕ_(5)-C₇₈

To determine the molecular structures of VSc₂C@C₈₀ (II) and VSc₂C@C₇₈, we used decapyrrylcorannulene (DPC) as a host to co-crystallize them[24–28]. Black single co-crystals were both obtained and used for single-crystal X-ray diffraction study, accomplishing unambiguous determination of their molecular structures as VSc₂C@_D₅ₕ_(6)-C₈₀ and VSc₂C@_D₃ₕ_(5)-C₇₈ (see Supplementary Tables 2–4 for the detailed crystallographic data and discussion). Figure 2a, b exhibit the molecular structures of these two μ₃-CCFs together with DPC molecules within VSc₂C@_D₅ₕ_(6)-C₈₀·2(DPC)·4(C₇H₈) and VSc₂C₂@_D₃ₕ_(5)-C₇₈·2(DPC)·4(C₇H₈) co-crystals, revealing that the DPC molecules adopt V-shape configuration and embrace two fullerene molecules. Such a stoichiometric ratio of 1:2 is quite different from the 1:1 ratio in the co-crystals of endohedral fullerenes with the commonly used Niᴵᴵ(OEP) (OEP = octaethylporphyrin) host, suggesting their difference in host-guest interactions[29–33]. To date, all crystallographically determined μ₃-CCFs, including TiM₂C@_Iₕ_(7)-C₈₀ (M = Sc[13], Tb[14], Dy[16], Lu[11]), USc₂C@_Iₕ_(7)-C₈₀ (ref. 18), and VSc₂C@_Iₕ_(7)-C₈₀ (ref. 19), are based on _Iₕ_(7)-C₈₀ cage. Although non-_Iₕ_-symmetry cages such as TiSc₂C@_D₅ₕ_(6)-C₈₀, TiDy₂C@_D₅ₕ_(6)-C₈₀ and TiSc₂C@_D₃ₕ_(5)-C₇₈ have been isolated, none of them were crystallographically characterized[12,13]. Therefore, VSc₂C@_D₅ₕ_(6)-C₈₀ and VSc₂C@_D₃ₕ_(5)-C₇₈ represent the first non-_Iₕ_-symmetry μ₃-CCFs with the molecular structures unambiguously determined by X-ray crystallography.

The fullerene cages of VSc₂C@_D₅ₕ_(6)-C₈₀ and VSc₂C@_D₃ₕ_(5)-C₇₈ are disordered in two orientations. Similar to the cases of VSc₂C@_Iₕ_(7)-C₈₀ and other EMFs[19,34], the encapsulated V/Sc atoms within VSc₂C cluster exhibit obvious disorders due to the thermal vibration, whereas

the central carbon atoms are fully ordered. Although the encapsulated VSc₂C clusters within VSc₂C@_D₅ₕ_(6)-C₈₀ and VSc₂C@_D₃ₕ_(5)-C₇₈ are both disordered in four orientations, V and Sc atoms can be distinguished according to a comparison of the R1/wR2 values obtained from different conformations of the encapsulated VSc₂C cluster combined with DFT calculations (see Supplementary Tables 5, 6 and Figs. 5–7 for details). Close-up views of the molecular structures of VSc₂C@_D₅ₕ_(6)-C₈₀ and VSc₂C@_D₃ₕ_(5)-C₇₈ with only major orientation of the fullerene cage and the major site of VSc₂C cluster are shown in Fig. 2c, d and Supplementary Fig. 8. For VSc₂C@_D₅ₕ_(6)-C₈₀, the V atom lies at the pentagon-hexagon conjunction, while the two Sc atoms are beneath the pentagon-hexagon-hexagon conjunctions. Upon decreasing the cage size to _D₃ₕ_(5)-C₇₈, V atom is beneath the hexagon-hexagon-hexagon conjunction and two Sc atoms both reside under the center of hexagon. Despite of the difference on the locations of metal atoms, the VSc₂C clusters within VSc₂C@_D₅ₕ_(6)-C₈₀ and VSc₂C@_D₃ₕ_(5)-C₇₈ both keep the planar triangle geometry since the sum of metal-carbon-metal angles is close to 360°. This feature resembles those of VSc₂C@_Iₕ_(7)-C₈₀ (ref. 19) and other crystallographically determined MSc₂C@_Iₕ_(7)-C₈₀ (M = Ti[13], U[18]) μ₃-CCFs.

The crystallographic characterizations of VSc₂C@_D₅ₕ_(6)-C₈₀ and VSc₂C@_D₃ₕ_(5)-C₇₈ facilitate analyses of V-C and Sc-C bonding natures. As illustrated in Fig. 2c, d, the lengths of V-C bond within VSc₂C@_D₅ₕ_(6)-C₈₀ and VSc₂C@_D₃ₕ_(5)-C₇₈ are 1.885(6) Å and 1.867(8) Å, respectively, which are comparable to that within VSc₂C@_Iₕ_(7)-C₈₀ (1.877(5) Å)[19] and thus can be assigned to V=C double bond. Interestingly, this feature differs from those of VSc₂N@_Iₕ_(7)-C₈₀ and VSc₂N@_D₅ₕ_(6)-C₈₀ NCFs in which V-N single bonds exist instead[22,23]. On the other hand, the lengths of Sc-C bonds in VSc₂C@_D₅ₕ_(6)-C₈₀ and VSc₂C@_D₃ₕ_(5)-C₇₈ are respectively 2.078(6)/2.186(5) Å and 2.054(10)/2.055(9) Å (Supplementary Table 7), which are very close to those observed in MSc₂C@_Iₕ_(7)-C₈₀ (M = V[19], Ti[13], U[18]), indicating that the two Sc–C bonds are single bonds. Notably, the metal-to-carbon distances in VSc₂C@_D₅ₕ_(6)-C₈₀ and VSc₂C@_Iₕ_(7)-C₈₀ are all slightly larger than that in VSc₂C@_D₃ₕ_(5)-C₇₈, suggesting stretching of VSc₂C cluster along with the cage expansion from C₇₈ to C₈₀. The V-C and Sc-C bonding

a

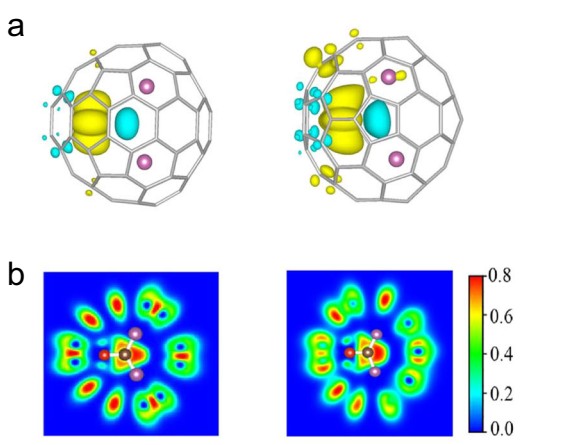

b

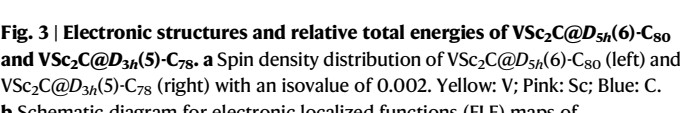

c

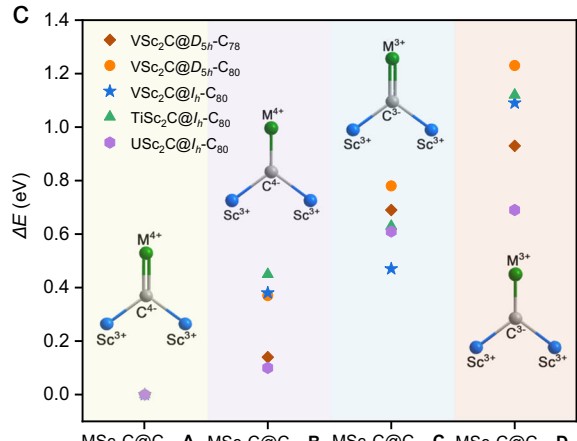

**Fig. 3 | Electronic structures and relative total energies of VSc$_2$C@$D_{5h}$(6)-C$_{80}$ and VSc$_2$C@$D_{3h}$(5)-C$_{78}$. a** Spin density distribution of VSc$_2$C@$D_{5h}$(6)-C$_{80}$ (left) and VSc$_2$C@$D_{3h}$(5)-C$_{78}$ (right) with an isovalue of 0.002. Yellow: V; Pink: Sc; Blue: C. **b** Schematic diagram for electronic localized functions (ELF) maps of VSc$_2$C@$D_{5h}$(6)-C$_{80}$ (left) and VSc$_2$C@$D_{3h}$(5)-C$_{78}$ (right) acting on the plane of the entrapped clusters. Red: V; Pink: Sc; Gray: C. **c** The relative total energies (ΔE, eV) of MSc$_2$C@C$_{2n}$ (M = V, Ti, U, 2n = 78, 80) μ$_3$-CCFs with different electronic configurations of MSc$_2$C cluster.

features observed in VSc$_2$C@$D_{5h}$(6)-C$_{80}$ and VSc$_2$C@$D_{3h}$(5)-C$_{78}$ as well as VSc$_2$C@$I_h$(7)-C$_{80}$ highly resemble those in the reported μ$_3$-CCFs TiM$_2$C@$I_h$(7)-C$_{80}$ (M = Sc[13], Tb[14], Dy[16], Lu[11]) and USc$_2$C@$I_h$(7)-C$_{80}$ (ref. 18), indicating the considerable structural similarity among μ$_3$-CCFs in terms of the existence of one double bond between the non-RE metal and the central carbon atom along with two RE metal-C single bonds. This stimulates us to propose a stabilization mechanism of μ$_3$-CCF as discussed below.

## DFT calculations of electronic configurations of VSc$_2$C@$D_{5h}$(6)-C$_{80}$ and VSc$_2$C@$D_{3h}$(5)-C$_{78}$

To investigate the electronic structures of VSc$_2$C@$D_{5h}$(6)-C$_{80}$ and VSc$_2$C@$D_{3h}$(5)-C$_{78}$ including the valence states of the encapsulated V atom and the interaction between V and C atoms, we carried out DFT calculations with the Vienna Ab-initio Simulation Package (VASP) at generalized gradient approximation (GGA) in the Perdew-Burke-Ernzerhof (PBE) levels[35]. According to DFT optimized molecular structures (Supplementary Fig. 9), the V-C bond lengths within VSc$_2$C@$D_{5h}$(6)-C$_{80}$ and VSc$_2$C@$D_{3h}$(5)-C$_{78}$ are 1.850 Å and 1.810 Å, respectively, while the Sc-C distances are 2.15/2.15 Å and 2.20/2.22 Å. These theoretical predictions agree well with the crystallographical results discussed above. To explore electronic structures of the VSc$_2$C@$D_{5h}$(6)-C$_{80}$ and VSc$_2$C@$D_{3h}$(5)-C$_{78}$, we calculated the spin-resolved molecular levels and plot the spatial distributions of several frontier molecular orbitals (Supplementary Fig. 10). It is clear that two molecules possess the spin-polarized ground states. Most frontier molecular orbitals are delocalized on the whole carbon cages, while there are also some localized molecular orbitals, mainly contributed by the inner VSc$_2$C cluster. For example, for the spin-up electrons, the single occupied molecular orbitals (SOMO-1) of VSc$_2$C@$D_{5h}$(6)-C$_{80}$ and SOMO-3 of VSc$_2$C@$D_{3h}$(5)-C$_{78}$ mainly localized around the V atom, according to the percentage of the V occupations in the majority-orbital composition. These results imply that there is one unpaired electron for V atom in VSc$_2$C@$D_{5h}$(6)-C$_{80}$ and VSc$_2$C@$D_{3h}$(5)-C$_{78}$, leading to the doublet ground states. The total magnetic moments of VSc$_2$C@$D_{5h}$(6)-C$_{80}$ and VSc$_2$C@$D_{3h}$(5)-C$_{78}$ are predicted to be about 1.02 and 0.98 μ$_B$, respectively. Figure 3a illustrates the spin-density spatial distributions of two molecules. Clearly, the spin-density mainly localized around the V atom, indicating that the V atom contributes mainly to the total magnetic moment. The atomic magnetic moment of V atom is about 1.15 and 1.05 μ$_B$ in VSc$_2$C@$D_{5h}$(6)-C$_{80}$ and VSc$_2$C@$D_{3h}$(5)-C$_{78}$, respectively. Note that, in these two molecules the

V atom antiferromagnetically couples with the neighboring C atom. The calculated partial density of states (DOS) for the V atom's spin-split d orbitals of VSc$_2$C@$D_{5h}$(6)-C$_{80}$ and VSc$_2$C@$D_{3h}$(5)-C$_{78}$ display different distributions for the majority and minority electrons (Supplementary Fig. 11). The molecular magnetic moments with one unpaired electron are mainly contributed by the 3$d_{z^2}$ orbitals of the V atom. At the same time, the 3$d_{xy}$ and 3$d_{yz}$ orbitals give the non-negligible contributions to total magnetic moments of VSc$_2$C@$D_{5h}$(6)-C$_{80}$ and VSc$_2$C@$D_{3h}$(5)-C$_{78}$, respectively. The electron configuration of the V atom is [Ar]3$d^3$4$s^2$, and one unpaired electron means the loss of four valence electrons from the V atom, resulting in a valence state of V$^{4+}$ within both μ$_3$-CCFs.

The V-C bonding type within VSc$_2$C@$D_{5h}$(6)-C$_{80}$ and VSc$_2$C@$D_{3h}$(5)-C$_{78}$ can be analyzed by the electronic localized function (ELF) maps, shown in Fig. 3b. Obviously, the interactions between V and C atoms are similar for two molecules. The visible electronic distributions are found on V atom, implying covalent interactions between V and C atoms. This is similar to the case of VSc$_2$C@$I_h$(7)-C$_{80}$ (ref. 19). Note that the electronic distribution on the V atom in VSc$_2$C@$D_{5h}$(6)-C$_{80}$ is slightly less than that in VSc$_2$C@$D_{3h}$(5)-C$_{78}$. This observation is consistent with their difference on the predicted V-C distances (1.850 and 1.810 Å in VSc$_2$C@$D_{5h}$(6)-C$_{80}$ and VSc$_2$C@$D_{3h}$(5)-C$_{78}$, respectively). The different electronic distributions in these two molecules can be understood by the analysis of the Bader charges, since the electronic transfer from the V atom to the outer cage within VSc$_2$C@$D_{5h}$(6)-C$_{80}$ (1.33 e) is larger than that within VSc$_2$C@$D_{3h}$(5)-C$_{78}$ (1.30 e). Combining the analyses of the spin density, the frontier molecular orbitals, partial density of states, and ELF maps, a + 4 formal valence state of the encapsulated V and covalent interactions for V = C double bonds are revealed for both VSc$_2$C@$D_{5h}$(6)-C$_{80}$ and VSc$_2$C@$D_{3h}$(5)-C$_{78}$ μ$_3$-CCFs. Noteworthy, The proposed V$^{4+}$ configuration within VSc$_2$C@$I_h$(7)-C$_{80}$ and VSc$_2$C@$D_{5h}$(6)-C$_{80}$ μ$_3$-C-CFs is obviously different to the V$^{3+}$ state within the reported VSc$_2$N@$I_h$(7)-C$_{80}$ and VSc$_2$N@$D_{5h}$(6)-C$_{80}$ NCFs[22,23]. Hence, the valence state of V can be steered via simply altering the non-metal atom within the encapsulated metal cluster.

## Supplemental Octet Rule for μ$_3$-carbido ligand in μ$_3$-CCFs

Due to the multihapto nature of the carbon cage, EAN rule is inapplicable for μ$_3$-CCFs. To understand the peculiar formation of the μ$_3$-carbido ligand in μ$_3$-CCFs, we carried out a systematic DFT study on the stabilities of VSc$_2$C@$I_h$(7)-C$_{80}$, VSc$_2$C@$D_{5h}$(6)-C$_{80}$ and

$VSc_2C@D_{3h}(5)-C_{78}$ with different electronic configurations based on M = C/M-C bonds with $M^{4+}/M^{3+}$ valence states, combined with those of the reported $MSc_2C@I_h(7)-C_{80}$ (M=Ti, U) $\mu_3$-CCFs as the representative members based on other non-rare earth metals of Ti, U. As seen from Fig. 3c, clearly $[V^{4+}(Sc^{3+})_2C^{4-}]^{6+}@[I_h(7)-C_{80}]^{6-}$, $[V^{4+}(Sc^{3+})_2C^{4-}]^{6+}@[D_{5h}(6)-C_{80}]^{6-}$ and $[V^{4+}(Sc^{3+})_2C^{4-}]^{6+}@[D_{3h}(5)-C_{78}]^{6-}$ based on $M^{4+}=C^{4-}$ double bond are the most stable configurations with the lowest total energies (see also Supplementary Table 8). Altering the $M^{4+}=C^{4-}$ double bond to $M^{4+}-C^{4-}$ single bond/$M^{3+}=C^{3-}$ double bond/$M^{3+}-C^{3-}$ single bond results in increased total energies and consequently less stable configurations. Similar results are obtained for $MSc_2C@I_h(7)-C_{80}$ (M = Ti, U) $\mu_3$-CCFs. Therefore, for $MSc_2C@C_{2n}$ (M = V, Ti, U; 2n = 80, 78) $\mu_3$-CCFs, $M^{4+}/Sc^{3+}$ cations and the central $C^{4-}$ anion are needed, affording one M = C double bond along with two Sc-C single bonds. The entire $MSc_2C$ cluster then transfers six electrons to the outer fullerene cage, enabling stable $\mu_3$-CCF molecule. In this way, the central $C^{4-}$ anion exhibits an eight-electron configuration. So far, there are sixteen $\mu_3$-CCFs in total have been reported, we further calculated the relative energies of the other eleven $\mu_3$-CCFs with different configurations. We find that, similar to the cases of $MSc_2C@C_{2n}$ (M = V, Ti, U, 2n = 78, 80), the configuration bearing a central carbon with an eight-electron configuration is the most stable structure for all $\mu_3$-CCFs (Supplementary Table 9). Based on these results, we propose a supplemental Octet Rule, that the central $\mu_3$-carbido ligand prefers to have eight electrons in the valence shell so as to be stabilized in $\mu_3$-CCF. In fact, the classic Octet Rule proposed by Lewis in 1916 has been commonly used for covalent compounds[36,37], but never been used for metal carbido complexes. In this work, we manage to extend it to $\mu_3$-CCFs as a special type of metal carbido complexes, and succeed in interpreting the peculiar formation of the $\mu_3$-carbido ligand within $\mu_3$-CCFs. Noteworthy, this supplemental Octet Rule is also applicable for the conventional binuclear metal carbido complexes containing a carbido bridge such as $L_nM = C = ML_n$ and $L_nM \equiv C-M'L_n$, but it is not always valid for $\mu_3$-carbido ligand within the conventional trinuclear metal carbido complexes[4–8]. Furthermore, in the metal carbido complexes bearing $\mu_5$- and $\mu_6$-carbido ligands, this Rule is not applicable any more because the central carbon atom is coordinated with more than four metals and thus is a hypervalency carbon[9,10]. For the well-known metal carbido complex $Fe_7MoS_9C$ as the active site of nitrogenase, although a central $C^{4-}$ anion also exists, it violates the supplemental Octet Rule because the central $C^{4-}$ anion bonds with six iron atoms to form a $Fe_6C$ core[2,38].

According to this supplemental Octet Rule, the necessity of involving a four-valency non-RE metal for the formation of $MSc_2C@C_{2n}$ $\mu_3$-CCFs can be easily understood, since a $M^{4+}$ cation is demanded to accomplish M = C double bond whereas +4 valence state is generally not preferable for the RE metals[39]. This also accounts for the absence of Sc-only $\mu_3$-CCF $Sc_3C@C_{80}$ under our synthesis condition, which is theoretically predicted very recently as an unstable free radical with one unpaired electron on the cage derived from the formal five-electron transfer[40], since $Sc^{4+}$ cation is hardly accessible. Interestingly, once the one deficient electron in the outer cage of unstable $Sc_3C@C_{80}$ is compensated by encapsulating one hydrogen atom, $Sc_3CH@C_{80}$ with an electronic configuration of $[(Sc^{3+})_3C^{4-}H^+]^{6+}@[C_{80}]^{6-}$ forms and the supplemental Octet Rule is also satisfied for the central $C^{4-}$ anion[41,42]. Likewise, $V_2ScC@C_{80}$ based on two $V^{4+}$ cations would violate the Octet Rule and thus seems also impossible. Therefore, this supplemental Octet Rule may be used as a simple guide for design of $\mu_3$-CCFs, offering opportunity to encapsulate other non-rare earth metals into fullerene cage.

It is intriguing to examine the applicability of this supplemental Octet Rule in other types of clusterfullerenes. $VSc_2N@I_h(7)-C_{80}$ NCF as an analogous V-containing trimetallic clusterfullerene is considered. Interestingly, upon changing the central non-metal atom from carbon to nitrogen, $[V^{3+}(Sc^{3+})_2N^{3-}]^{6+}@[I_h(7)-C_{80}]^{6-}$ based on $V^{3+}-N^{3-}$ single bond becomes the most stable configuration (see Supplementary Table 10), as experimentally confirmed[22]. Since N atom has five valence electrons, upon formation of $VSc_2N@I_h(7)-C_{80}$ NCF, $V^{3+}/Sc^{3+}$ cations along with three V-N/Sc-N single bonds exist, rendering an eight-electron configuration for the central $N^{3-}$ anion as well. Therefore, this supplemental Octet Rule is also applicable for NCF.

### Electronic properties of $VSc_2C@D_{5h}(6)-C_{80}$ and $VSc_2C@D_{3h}(5)-C_{78}$

In order to probe the effect of the central nonmetal atom (C/N) on the electronic properties of $\mu_3$-CCF and NCF, we carried out UV-vis-NIR spectroscopic and electrochemical characterizations. Figure 4a compares the UV-vis−NIR absorption spectra of $VSc_2C@D_{5h}(6)-C_{80}$ and $VSc_2C@D_{3h}(5)-C_{78}$ dissolved in toluene (see Supplementary Fig. 12 and Table 11 for their characteristic absorption data along with analogous $\mu_3$-CCFs and NCFs). $VSc_2C@D_{5h}(6)-C_{80}$ exhibits a broad absorption peak at 447 nm and a minor shoulder peak at 383 nm, and the overall spectral feature looks similar to that of $VSc_2N@D_{5h}(6)-C_{80}$ but quite different from that of $VSc_2C@I_h(7)-C_{80}$[22,23]. This is understandable since the outer fullerene cages of $VSc_2C@D_{5h}(6)-C_{80}$ and $VSc_2N@D_{5h}(6)-C_{80}$ are same while $\pi − \pi^*$ transitions of the fullerene cage predominantly determines the electronic absorptions of EMFs[34]. For $VSc_2C@D_{3h}(5)-C_{78}$, two intense absorption peaks at 462 and

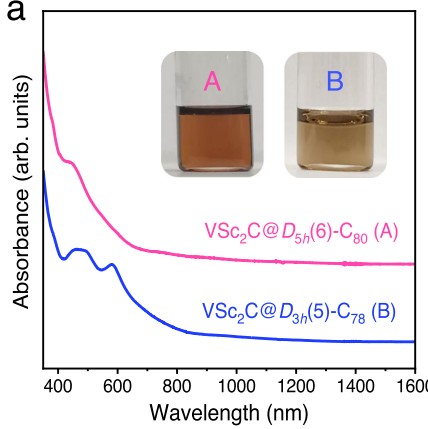
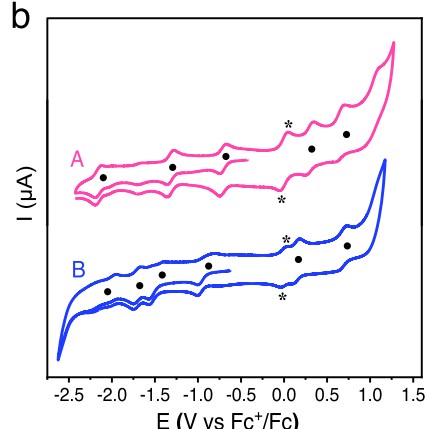

**Fig. 4 | Electronic properties of $VSc_2C@D_{5h}(6)-C_{80}$ and $VSc_2C@D_{3h}(5)-C_{78}$. a** UV-vis−NIR spectra of $VSc_2C@D_{5h}(6)-C_{80}$ and $VSc_2C@D_{3h}(5)-C_{78}$ dissolved in toluene. Insets: Photographs of their corresponding solutions in toluene. **b** Cyclic voltammograms of $VSc_2C@D_{5h}(6)-C_{80}$ and $VSc_2C@D_{3h}(5)-C_{78}$ in $o$-DCB solution with ferrocene (Fc) as the internal standard under different scan regions. Scan rate: 100 mV/s, $TBAPF_6$ as supporting electrolyte. The half-wave potentials ($E_{1/2}$) of each redox step are marked with a solid dot to aid comparison. The asterisk labels the oxidation and reduction peaks of ferrocene.

581 nm are observed, resembling $Sc_3N@D_{3h}(5)\text{-}C_{78}$ despite of some shifts of the absorption peaks due to the discrepancy on the encapsulated cluster. According to the absorption spectral onsets of 1490 and 1430 nm for $VSc_2C@D_{5h}(6)\text{-}C_{80}$ and $VSc_2C@D_{3h}(5)\text{-}C_{78}$, their optical bandgaps ($\Delta E_{\text{gap, optical}}$) are determined to be 0.83 and 0.87 eV, respectively, which are comparable to that of $VSc_2C@I_h(7)\text{-}C_{80}$ (0.88 eV)[19].

Although few $D_{5h}(6)\text{-}C_{80}$- and $D_{3h}(5)\text{-}C_{78}$-based $\mu_3$-CCFs like $TiSc_2C@D_{5h}(6)\text{-}C_{80}$, $TiDy_2C@D_{5h}(6)\text{-}C_{80}$ and $TiSc_2C@D_{3h}(5)\text{-}C_{78}$ were isolated before, their electrochemical properties have never been investigated yet. Figure 4b presents cyclic voltammograms of $VSc_2C@D_{5h}(6)\text{-}C_{80}$ and $VSc_2C@D_{3h}(5)\text{-}C_{78}$ measured in $o$-dichlorobenzene ($o$-DCB) with tetrabutylammonium hexafluorophosphate ($TBAPF_6$) as supporting electrolyte (see Supplementary Fig. 13 for cyclic voltammograms in different scanning regions), and their characteristic redox potentials along with the analogous NCFs are summarized in Supplementary Table 12. For $VSc_2C@D_{3h}(5)\text{-}C_{78}$, two reversible oxidation steps with half-wave potentials ($E_{1/2}$) at 0.14 and 0.69 V in the anodic region are observed, and the first oxidation potential is more negative than that measured for $Sc_3N@D_{3h}(5)\text{-}C_{78}$ while the second one is close to that of $Sc_3N@D_{3h}(5)\text{-}C_{78}$ (ref. 40). In the cathodic region, $VSc_2C@D_{3h}(5)\text{-}C_{78}$ shows quite different reductive behavior compared to $Sc_3N@D_{3h}(5)\text{-}C_{78}$ in terms of number of the reduction steps: $VSc_2C@D_{3h}(5)\text{-}C_{78}$ exhibits two irreversible reduction steps and two reversible reduction steps, while there are only two irreversible reduction steps for $Sc_3N@D_{3h}(5)\text{-}C_{78}$ (ref. 43). In particular, the first reduction potential ($^{\text{red}}E_1$) of $VSc_2C@D_{3h}(5)\text{-}C_{78}$ is positively shifted by 650 mV relative to that of $Sc_3N@D_{3h}(5)\text{-}C_{78}$. The more positive $^{\text{red}}E_1$ and more negative first oxidation potential ($^{\text{ox}}E_1$) of $VSc_2C@D_{3h}(5)\text{-}C_{78}$ result in a much smaller electrochemical gap (1.05 eV) than that of $Sc_3N@D_{3h}(5)\text{-}C_{78}$ (1.77 eV). Similar phenomenon is observed for $VSc_2C@D_{5h}(6)\text{-}C_{80}$, which exhibits two reversible oxidation steps with $E_{1/2}$ at 0.30 and 0.66 V and three reversible reduction steps with $E_{1/2}$ at −0.70, −1.31, and −2.16 V. The $^{\text{ox}}E_1$ and $^{\text{ox}}E_2$ values are both more negative than those of $VSc_2C@I_h(7)\text{-}C_{80}$ and $VSc_2N@D_{5h}(6)\text{-}C_{80}$. In addition, the reductive behavior of $VSc_2C@D_{5h}(6)\text{-}C_{80}$ is more different with $VSc_2N@D_{5h}(6)\text{-}C_{80}$ which shows four reversible reduction steps instead, and the electrochemical gap of $VSc_2C@D_{5h}(6)\text{-}C_{80}$ (1.00 eV) is smaller than that of $VSc_2N@D_{5h}(6)\text{-}C_{80}$ (1.20 eV)[23]. Therefore, the encapsulated cluster especially the central nonmetal atom affects sensitively the electronic properties of the trimetallic clusterfullerene.

## Discussion

In summary, three V-based $\mu_3$-CCFs, namely $VSc_2C@I_h(7)\text{-}C_{80}$, $VSc_2C@D_{5h}(6)\text{-}C_{80}$ and $VSc_2C@D_{3h}(5)\text{-}C_{78}$, are successfully synthesized and isolated, among them the latter two represent the first crystallographically determined non-$I_h$-symmetry $\mu_3$-CCFs. Their molecular structures are determined unambiguously by single-crystal X-ray diffraction, revealing the existence of V = C double bonds and high valence state of V (+4). This differs from those of $VSc_2N@I_h(7)\text{-}C_{80}$ and $VSc_2N@D_{5h}(6)\text{-}C_{80}$ NCFs in which V-N single bonds and $V^{3+}$ valence state exist. The encapsulated cluster especially the central nonmetal atom affects sensitively the electronic properties of $\mu_3$-CCF and NCF trimetallic clusterfullerenes. On the basis of a systematic DFT study on the stabilities of all reported sixteen $\mu_3$-CCFs, a supplemental Octet Rule is proposed, that the central $\mu_3$-carbido ligand prefers to have eight electrons in the valence shell so as to be stabilized in $\mu_3$-CCF. The applicability of this supplemental Octet Rule in other types of clusterfullerenes is exemplified by $VSc_2N@I_h(7)\text{-}C_{80}$ NCF. By applying the classic Octet Rule to $\mu_3$-CCFs as the simplest metal carbido complexes, we establish a rule beyond the EAN rule commonly used during the past century, offering new insight into the stability criteria of multinuclear clusterfullerenes containing single-atom-ligand.

## Methods

### Synthesis and isolation of $VSc_2C@I_h(7)\text{-}C_{80}$, $VSc_2C@D_{5h}(6)\text{-}C_{80}$ and $VSc_2C@D_{3h}(5)\text{-}C_{78}$

$VSc_2C@I_h(7)\text{-}C_{80}$, $VSc_2C@D_{5h}(6)\text{-}C_{80}$ and $VSc_2C@D_{3h}(5)\text{-}C_{78}$ were synthesized in a Krätschmer-Huffman generator by vaporizing composite graphite rods containing a mixture of $Sc_2O_3$, VC and graphite powder (the molar ratio of Sc:V:C = 1:1:15) as the raw material with the addition of 200 mbar He. The produced soot was collected and Soxhlet-extracted by $CS_2$ for 24 h. The resulting brown-yellow solution was distilled to remove $CS_2$, and then immediately redissolved in toluene and subsequently passed through a 0.2 μm Telflon filter (Sartorius AG, Germany) for HPLC separation. I $VSc_2C@I_h(7)\text{-}C_{80}$, $VSc_2C@D_{5h}(6)\text{-}C_{80}$ and $VSc_2C@D_{3h}(5)\text{-}C_{78}$ were isolated by three/four-step HPLC (LC-9104, Japan Analytical Industry) as described in details in Supplementary Figs. 1–4. The relative abundance of the products is shown in the Supplementary Table 1. The purity of the isolated $VSc_2C@D_{5h}(6)\text{-}C_{80}$ and $VSc_2C@D_{3h}(5)\text{-}C_{78}$ were checked by HPLC and LD-TOF MS (Biflex III, Bruker Daltonics Inc., Germany).

### Spectroscopic and electrochemical study

UV-vis–NIR spectra of $VSc_2C@D_{5h}(6)\text{-}C_{80}$ and $VSc_2C@D_{3h}(5)\text{-}C_{78}$ dissolved in toluene were recorded on a UV−vis−NIR 3600 spectrometer (Shimadzu, Japan) using a quartz cell of 1 mm layer thickness and 1 nm resolution. Electrochemical study of $VSc_2C@D_{5h}(6)\text{-}C_{80}$ and $VSc_2C@D_{3h}(5)\text{-}C_{78}$ were performed in $o$-dichlorobenzene ($o$-DCB, anhydrous, 99%, Aldrich). The supporting electrolyte was tetrabutylamonium hexafluorophosphate ($TBAPF_6$, puriss. electrochemical grade, Fluka) which was dried under pressure at 340 K for 24 h and stored in glovebox prior to use. Cyclic voltammogram experiments were performed with a CHI 660D potentiostat (CHI Instrument, USA) at room temperature. A standard three-electrode arrangement of a platinum (Pt) disc as working electrode, a platinum wire as counter electrode, and a silver wire as an auxiliary electrode was used. In a comparison experiment, ferrocene (Fc) was added as the internal standard and all potentials are referred to Fc/Fc$^+$ couple.

### X-ray crystallographic study

Crystal growths of $VSc_2C@D_{5h}(6)\text{-}C_{80}$ and $VSc_2C@D_{3h}(5)\text{-}C_{78}$ were accomplished by slow evaporation from mixed solutions of purified sample and DPC in toluene, and small black crystals suitable for X-ray crystallographic study were obtained after two weeks. The crystallographic characterization was performed in beamline station BL17B at Shanghai Synchrotron Radiation Facility. The structure was refined using all data (based on F$^2$) by SHELXL 2015 (ref. 44) within OLEX2 (ref. 45). A summary of the crystallographic data is listed in Supplementary Table 2. The ORTEP-style illustration with probability ellipsoids and notes on CheckCif file B-level alerts are shown in Supplementary Fig. 14 and Supplementary Table 13.

### Computations

In our calculations, the geometrical structures and electronic properties were explored by performing spin-polarized density functional theory (DFT) methods implemented in the Vienna Ab-initio Simulation Package (VASP). The generalized gradient approximation (GGA) in the Perdew-Burke-Ernzerhof (PBE) form was adopted to describe the exchange and correlation energy[35]. The energy cutoff of 400 eV was selected for the plane wave expansion and an automatic k-point mesh (1 × 1 × 1) was generated with a Gamma-centered grid.

### Data availability

Crystallographic data of the structures reported in this Article have been deposited in the Cambridge Crystallographic Data Center (CCDC), under deposition numbers 2209496 ($VSc_2C@D_{3h}(5)\text{-}C_{78}\cdot2(DPC)\cdot3(C_7H_8)$), 2038584 ($VSc_2C@D_{5h}(6)\text{-}C_{80}\cdot2(DPC)\cdot4(C_7H_8)$) and 2038583 ($VSc_2C@I_h(7)\text{-}C_{80}\cdot2(DPC)\cdot3(C_7H_8)$). Copies of the data

can be obtained free of charge via https://www.ccdc.cam.ac.uk/structures/. All other data that support the findings of this study are available from the Supplementary Information and/or from the corresponding author upon request.

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

## Acknowledgements

S.Y. thanks National Natural Science Foundation of China (51925206, U1932214), the Fundamental Research Funds for the Central Universities (20720220009) and the Strategic Priority Research Program of the Chinese Academy of Sciences (XDB0450301). Q.L. thanks National Natural Science Foundation of China (21873088, 91961113) and Innovation Program for Quantum Science and Technology (2021ZD0303306). Y.T. and S.X. thank National Natural Science Foundation of China (92061103, 92061204 and 22101241). We acknowledge the staff in the BL17B beamline of the National Facility for Protein Science Shanghai (NFPS) at the Shanghai Synchrotron Radiation Facility for assistance during data collection.

## Author contributions

S.Y. conceived and designed this research. R.G. synthesized and separated the fullerene samples and conducted characterizations. J.H. and Q.L. carried out DFT calculations. J.X. helped with separation of samples. M.C. and P.D. helped with X-ray crystallographic measurements and analysis. Y.T. and S.X. provide decapyrrylcorannulene. R.G., J.H., Q.L., Y.T., S.X. and S.Y. co-wrote the paper, and all the authors commented on it.

## Competing interests

The authors declare no competing interests.
