## [Peer Review File · Nature Communications]

A stabilization rule for metal carbido cluster bearing μ_3 -carbido single-atom-ligand encapsulated in carbon cageReviewers' Comments:

Reviewer #1:

Remarks to the Author:

This article reports the synthesis, isolation and structural characterization of two new endohedral fullerenes. The synthesis and isolation are rather routine but time consuming.

The structural characterization by X-ray crystallography is problematic. The authors must report esds for bond distances, angle and site occupancies if refined. They didn't. Differentiation between Sc and V crystallographically is challenging because of their similar atomic numbers and similar scattering. Inside the fullerene there are many disordered sites for the metal ions. How did the authors choose to assign sites to V or Sc? The assumptions they may have made need to be specified. It is possible that Sc and V may occupy the same site. How can one tell? Site occupancies and disorder are poorly handled in the text itself and much is missing the SI. A full list of Sc-C(carbide) and V-C(carbide) distances should be available either in the text or SI and those data should be compared with Sc-N and V-N distances in relevant $M_3N@C_{80}$ and $M_3N@C_{78}$ compounds. Those comparisons would allow the reader to better evaluate the claims made by the authors. The drawings of the two new endohedrals should clearly identify the principle C_5 and C_3 axes in each. In C_{78} , where are the metal sites in regard to the C_3 axis? Figure 2 is inadequate, since it is much too small and crowded.

A second aspect of the paper is the discussion of an "expanded Octet Rule" for endohedrals containing a central carbide ion. I did not find this topic to have much merit. Generally the term "expanded octet" was used for situations where valence bond theory suggested the participation of d orbitals for elements in the second or lower in the periodic table. Since carbon only has s and p electrons available in the valence shell, the octet rule pertains to carbon period! If one considered the central carbon as a carbide ion, then having any number of metal ions surrounding it presents no problems. The authors fail to make a convincing case that they can add anything of substance to our understanding of carbide-containing endohedrals in this paper.

I cannot recommend this article for publication. The crystallography appears to be flawed. The ideas involved in the "expanded Octet Rule" do not enlarge our understanding of carbide containing clusters. The article will have little impact on our understanding of chemical bonding.

Reviewer #2:

Remarks to the Author:

The authors present some newly synthesised cage compounds. I am not from the field but these compounds are certainly interesting and challenging to synthesise. It is noted that the newly synthesised compound has an octet configuration on the central $\mu\text{-3}$ carbon and this is used in an attempt to derive an expanded octet rule. Generally speaking, the compound synthesised is interesting. However, I don't believe that from the evidence presented one can really derive a general rule. The paper is generally well-presented but there is a bit too much jargon to reach a general audience.

1. If the authors really derived a general expanded octet rule, then they should explain its applications and its limits. One compound, present in several different cages, can hardly be used to derive a new rule. Can the authors explain unexplained trends in the literature or make new predictions using their rule?

2. The cage compounds are described with their specific notation $V_{sc}2C@Ih(7)-C_{80}$, etc. It would be good, if possible, to reduce this kind of jargon and explain in words what is the significance of the different compounds discussed. Why is it important that the compound is not of Ih symmetry?

3. The authors should explain the level of theory used. Is PBE with a plane-wave basis really the optimal choice? Shouldn't one at least use a dispersion correction to model the cage?
4. The authors should explain how they really derive the electronic configuration of the metal ions.
5. The authors say that the "molecules possess the spin-split ground states." It should be explained what is meant by that. Spin-orbit coupling? Or is it because broken symmetry DFT is used?

Reviewer #3:

Remarks to the Author:

Herein, the authors propose a new rule that would govern the stabilities of metal carbide complexes. After having synthesized, purified, and characterized 3 new vanadium-based μ_3 CCFs, the authors are seeking and proposing an explanation for their unique electron configuration. In their TiSc₂C clusters, they discuss an expanded octet from the high valence state of a Ti⁴⁺ encapsulated metal cluster. The authors iteratively blend experiments and DFT calculations to support their rationale for an expanded octet rule for the μ_3 -carbido ligand in μ_3 -VSc₂C based CCFs.

One of the novel features of this paper is the peculiar bonding of the encapsulated metal carbide complex. Even the authors themselves address the novelty perspective with their statement on page 12: "In fact, the classic Octet Rule proposed by Lewis in 1916 has been commonly used for covalent compounds, but never been used for metal carbido complexes."

Indeed, one of the intriguing features of endohedral metallofullerenes is the opportunity given to us by nature for creating exotic bonding types engendered by the encapsulated cluster within the carbon fullerene cage housing. With a reputation for offering unique and uncommon bonding types stemming from endohedral clusters, such an expanded octet proposition is indeed credible. In short, the credibility of the author's proposal of an expanded Octet Rule for a μ_3 -carbido single-atom-ligand stabilized via the carbon cage is enhanced by such a unique molecular architecture as found in endohedral metallofullerenes.

Strengths:

The authors have synthesized, purified, and characterized 3 new species not known to exist, much less purified. The X-ray crystal structures were essential. Combined with DFT calculations, the authors have a scientifically sound basis for their proposed explanation of the expanded octet.

The results are valid and sound. Indeed, the amount of data and methodology from a variety of sources is very extensive. After reading carefully the manuscript and even the supplementary information, I do not find any glaring weak areas, and I have no suggested additional improvements. The iterative blend of DFT support and experimental data fortifies the credibility and soundness of the proposed expanded octet.

Conclusion:

The authors' contribution to Nature Chemistry has a high significance and novelty factor. I believe there to be interest to multiple fields of science. Indeed, the broad interest is address well on page 2 of the introduction. Moreover, there is an impressive amount of scientific support for their proposal of an expanded Octet Rule for these VSc₂C endohedral metallofullerenes. In totality, my opinion is in support of publication in Nature Communications.

Point-by-point response to reviewer's comments:

Reviewer #1 (Remarks to the Author):

This article reports the synthesis, isolation and structural characterization of two new endohedral fullerenes. The synthesis and isolation are rather routine but time consuming.

We thank the Reviewer for the critical comments, which helped us to improve our manuscript.

Comment 1: *The structural characterization by X-ray crystallography is problematic. The authors must report esds for bond distances, angle and site occupancies if refined. They didn't.*

Answer: We thank the Reviewer for the carefulness on checking the X-ray crystallographic data. We did refine all parameters, but omitted them for simplicity in the original version. Following the Reviewer's suggestion, we added all esds for bond distances, angle, and site occupancies for VSc₂C@D_{5h}(6)-C₈₀ and VSc₂C@D_{3h}(5)-C₇₈ in Figure 2, the main text and SI (Supplementary Figures 6-7, Table 3, 4 and 7). Please note that, after double checking all crystallographic data, subtle changes on the parameters of VSc₂C@D_{3h}(5)-C₇₈ happened based on improved data.

Comment 2: *Differentiation between Sc and V crystallographically is challenging because of their similar atomic numbers and similar scattering. Inside the fullerene there are many disordered sites for the metal ions. How did the authors choose to assign sites to V or Sc? The assumptions they may have made need to be specified. It is possible that Sc and V may occupy the same site. How can one tell? Site occupancies and disorder are poorly handled in the text itself and much is missing the SI.*

Answer: We thank the Reviewer for the professional opinion. Indeed, based on crystallography theory, differentiation between Sc and V crystallographically (without specific requirement on how they are combined together) is challenging because of their similar atomic numbers and similar scattering. Fortunately, in our present work, because of the requirement of formation of planar triangular VSc₂C cluster following "Supplemental Octet Rule", specific electronic configuration of [V⁴⁺(Sc³⁺)₂C⁴⁺]⁶⁺@[C_{2n}]⁶⁻ and metal-carbon bondings (one double bond plus two single bonds, see Figure 3c) are required, enabling us to differentiate Sc and V unambiguously.

In our previous work (*Proc. Natl. Acad. Sci. U. S. A.* **119**, e2202563119 (2022)), we already presented details on how to crystallographically distinguish the Sc and V sites within $VSc_2C@I_h(7)-C_{80}$, by comparing R1 and wR2 values obtained from different configurations of the VSc_2C cluster combined with DFT calculations. Using the same method, we succeeded in distinguishing the Sc and V sites within $VSc_2C@D_{5h}(6)-C_{80}$ and $VSc_2C@D_{3h}(5)-C_{78}$. Taking $VSc_2C@D_{3h}(5)-C_{78}$ as an example, we describe in details how to distinguishing the Sc and V sites as follow.

First, it is necessary to determine all disorder sites of three encapsulated metals (one V and two Sc atoms) within $VSc_2C@D_{3h}(5)-C_{78}$, as shown in Fig. R1.

Fig. R1. Positions of the disordered metal sites and bond lengths between metal sites and the central carbon atom in $VSc_2C@D_{3h}(5)-C_{78}$.

According to Fig. R1, there are totally twelve metal sites in $VSc_2C@D_{3h}(5)-C_{78}$. It is observed that M1a to M1d sites possess comparable bond lengths to the central carbon atom (1.86-1.91 Å), which are distinctly shorter than those for other metal sites (1.99-2.12 Å). Because the disordered sites of the same metal atom normally have similar bond lengths, it is reasonable to assign M1a to M1d sites to the same metal atom (M1) with different site occupancies. In addition, the other eight metal sites are distributed on both sides of the M1 disordered sites (see Fig. R1). Because the disordered sites of each metal atom are generally confined in the vicinity of the major sites, the four metal sites located on each side can be attributed to the disordered sites of the same metal atom (M2/M3). Therefore, each metal atom has four disordered sites, and the total twelve metal sites can be divided into four orientations of the encapsulated cluster VSc_2C (differentiated by different bonding colors), which are depicted in Fig. R2. In different orientations, the corresponding bond lengths are comparable. Moreover, these

four orientations all adopt planar configurations (i. e. the sum of M1-C79-M2, M2-C79-M3, and M3-C79-M1 is close to 360°), and crystallographic results suggest that the disorder of each metal atom is derived from the rotation of planar VSc₂C cluster.

Fig. R2 (see also **Supplementary Figure 7, revised version**). Four orientations of VSc₂C cluster in VSc₂C@D_{3h}(5)-C₇₈.

Next, to assign M1, M2, M3, we carried out additional study on refining possible configurations of the VSc₂C cluster with different V/Sc sites within VSc₂C@D_{3h}(5)-C₇₈. Since there are two Sc atom within VSc₂C cluster, which should be identical, we optimized three possible conformations of the encapsulated VSc₂C cluster within VSc₂C@D_{3h}(5)-C₇₈ (**A**, **B**, **C**) by using DFT calculations (Table R1). Based on DFT calculation results, VSc₂C@D_{3h}(5)-C₇₈-**A** in which V possesses the shorter bond lengths to the central carbon atom (around 1.8 Å) has the lowest relative energy with the smallest R1 and wR2 values (12.05%/35.24%). Besides, R1 and wR2 values both increase when changing the disordered V sites to other sites with longer bond length of approximately 2 Å with the central carbon (see Table R1). Therefore, combining the R1/wR2 values with DFT calculations, we concluded that VSc₂C@D_{3h}(5)-C₇₈-**A** is the most stable one, in which the short V-C bond length is attributed to V=C double bond and the two Sc-C bonds are longer in the form of single bonds. As seen in Fig. R2, the V-C bond lengths and Sc-C bond lengths based on other minor sites (orientations **B-D**) also fall into the range of V=C double bond and Sc-C single bond, respectively. Overall,

because of the distinct difference between V=C bond lengths and Sc-C bond lengths, in our present work it is feasible to distinguish V and Sc sites since it is hard for Sc and V atoms to occupy the same metal site.

Table R1 (added as Supplementary Table 5). Comparison of three possible conformations of $VSc_2C@D_{3h}(5)-C_{78}$.

	$VSc_2C@D_{3h}(5)-C_{78}-A$	$VSc_2C@D_{3h}(5)-C_{78}-B$	$VSc_2C@D_{3h}(5)-C_{78}-C$
Molecular structure			Relative Energy (eV)	0	0.09	0.08
R1 (%)	12.05	12.17	12.09
wR2 (%)	35.24	35.51	35.35

The case of $VSc_2C@D_{5h}(6)-C_{80}$ is similar to that of $VSc_2C@D_{3h}(5)-C_{78}$. The disordered sites of metal atoms within $VSc_2C@D_{5h}(6)-C_{80}$ can also be divided into four orientations (see Supplementary Figure 6). Table R2 (added as new Supplementary Table 6) also reveals that $VSc_2C@D_{5h}(6)-C_{80}-A$ bearing the V=C double is the most stable conformation with the lowest relative energy and the smallest R1 and wR2 values (8.65%/26.34%).

Fig. R3 (Supplementary Figure 6, revised version). Four orientations of VSc₂C unit in VSc₂C@D_{5h}(6)-C₈₀ including the major (a) and minor (b, c, d) orientations.

Table R2 (added as Supplementary Table 6). Comparison of three possible conformations of VSc₂C@D_{5h}(6)-C₈₀.

	VSc ₂ C@D _{5h} (6)-C ₈₀ -A	VSc ₂ C@D _{5h} (6)-C ₈₀ -B	VSc ₂ C@D _{5h} (6)-C ₈₀ -C
Molecular structure			
Relative energy (eV)	0	0.22	0.21
R1 (%)	8.65	8.87	8.82
wR2 (%)	26.34	27.65	27.85

To clarify this issue, we added one sentence in the paragraph discussing the X-ray structure “Although the encapsulated VSc₂C clusters within VSc₂C@D_{5h}(6)-C₈₀ and VSc₂C@D_{3h}(5)-C₇₈ are both disordered in four orientations, V and Sc atoms can be distinguished according to a comparison of the R1/wR2 values obtained from different

conformations of the encapsulated VSc₂C cluster combined with DFT calculations (see Supplementary Tables 5-6 and Figs. 5-7 for details).” (see Page 7, lines 8-13, revised version) Besides, we added Tables 5 and 6 along with detailed analysis in Supplementary Information.

Comment 3: A full list of Sc-C (carbide) and V-C (carbide) distances should be available either in the text or SI and those data should be compared with Sc-N and V-N distances in relevant M₃N@C₈₀ and M₃N@C₇₈ compounds. Those comparisons would allow the reader to better evaluate the claims made by the authors.

Answer: We thank the Reviewer for the constructive suggestion. Following the Reviewer’s suggestions, we added a full list of V-C and Sc-C bond lengths as the new Supplementary Table 4.

Table R3 (Supplementary Table 4, revised version). The V-C and Sc-C bond lengths (Å) based on different V and Sc sites within VSc₂C@D_{5h}(6)-C₈₀, VSc₂C@D_{3h}(5)-C₇₈ and VSc₂C@I_h(7)-C₈₀.

	VSc ₂ C@D _{5h} (6)-C ₈₀		VSc ₂ C@D _{3h} (5)-C ₇₈		VSc ₂ C@I _h (7)-C ₈₀	
	V1-C81	1.885(6)	V1-C81	1.867(8)	V1-C81	1.877(5)
V-C	V1a-C81	1.923(5)	V1a-C81	1.909(7)	V1a-C81	1.874(5)
	V1b-C81	1.905(10)	V1b-C81	1.869(8)	V1b-C81	1.798(5)
	V1c-C81	1.812(11)	V1c-C81	1.884(9)		
	Sc1-C81	2.078(6)	Sc1-C81	2.054(10)	Sc1-C81	2.102(5)
Sc1-	Sc1a-C81	2.226(6)	Sc1a-C81	2.121(9)	Sc1a-C81	2.100(6)
	Sc1b-C81	2.264(7)	Sc1b-C81	2.097(8)	Sc1b-C81	2.147(7)
C	Sc1c-C81	2.162(9)	Sc1c-C81	2.059(12)		
	Sc2-C81	2.186(5)	Sc2-C81	2.055(9)	Sc2-C81	2.145(4)
Sc2-	Sc2a-C81	1.956(6)	Sc2a-C81	1.991(7)	Sc2a-C81	2.096(6)
	Sc2b-C81	1.920(9)	Sc2b-C81	2.077(8)	Sc2b-C81	2.207(5)
C	Sc2c-C81	2.176(8)	Sc2c-C81	2.022(10)		

In addition, we added the new Supplementary Table 7, in which a comparison of the V-C and Sc-C bond lengths within VSc₂C@D_{3h}(5)-C₇₈, VSc₂C@D_{5h}(6)-C₈₀, and VSc₂C@I_h(7)-C₈₀ with those of Sc-N and V-N bonds within VSc₂N@I_h(7)-C₈₀,

$V_2ScN@I_h(7)-C_{80}$, $VSc_2N@D_{5h}(6)-C_{80}$, $Sc_3N@D_{3h}(5)-C_{78}$, $Sc_3N@I_h(7)-C_{80}$, and $Sc_3N@D_{5h}(6)-C_{80}$ is given. Comparing the M-C bond lengths within $VSc_2C@C_{2n}$ and M-N bonds lengths within $VSc_2N@C_{2n}$ and $Sc_3N@C_{2n}$, clearly Sc-C single bonds are mostly longer than Sc-N single bonds due to the larger electronegativity of nitrogen atom than carbon atom, whereas V-C bond lengths are comparable to V-N bond lengths within $VSc_2N@I_h(7)-C_{80}$ and even smaller than those within $VSc_2N@D_{5h}(6)-C_{80}$ and $V_2ScN@I_h(7)-C_{80}$, confirming the double bond nature the V-C bonds.

Table R4 (Supplementary Table 7, revised version). The lengths (Å) of metal-carbon bonds within the major VSc_2C cluster in $VSc_2C@D_{3h}(5)-C_{78}$, $VSc_2C@D_{5h}(6)-C_{80}$ and $VSc_2C@I_h(7)-C_{80}$, compared with the lengths of metal-nitrogen bonds within the corresponding $M_3N@C_{80}$ and $M_3N@C_{78}$ counterparts.

	$VSc_2C@D_{3h}(5)-C_{78}$	$VSc_2C@D_{5h}(6)-C_{80}$	$VSc_2C@I_h(7)-C_{80}$
V-C	1.867(8)	1.885(6)	1.877(5)
Sc1-C	2.054(10)	2.078(6)	2.102(5)
Sc2-C	2.055(9)	2.186(5)	2.145(4)
	$VSc_2N@I_h(7)-C_{80}$	$VSc_2N@D_{5h}(6)-C_{80}$	$V_2ScN@I_h(7)-C_{80}$
V1-N	1.858(7)	1.900(5)	1.995(5)
V2-N	–	–	2.027(3)
Sc1-N	2.036(4)	2.071(2)	2.003(6)
Sc2-N	2.036(4)	2.071(2)	–
	$Sc_3N@D_{3h}(5)-C_{78}$	$Sc_3N@D_{5h}(6)-C_{80}$	$Sc_3N@I_h(7)-C_{80}$
Sc1-N	1.9998(10)	2.014(2)	1.9931(14)
Sc2-N	2.0106(10)	2.031(2)	2.0323(16)
Sc3-N	2.0111(10)	2.041(2)	2.0526(14)

Comment 4: The drawings of the two new endohedrals should clearly identify the principle C_5 and C_3 axes in each. In C_{78} , where are the metal sites in regard to the C_3 axis? Figure 2 is inadequate, since it is much too small and crowded.

Answer: We thank the Reviewer for the constructive suggestion. Following the Reviewer's suggestions, we plotted a new figure (Fig. R4, added as new Supplementary Figure 8) to identify the principle C_5 and C_3 axes in $VSc_2C@D_{5h}(6)-C_{80}$ and

$VSc_2C@D_{3h}(5)-C_{78}$. For $VSc_2C@D_{3h}(5)-C_{78}$, it can be seen that the C_3 axis lies on the plane of VSc_2C cluster (see Fig. R4, c and d).

Considering the comment of Reviewer, we deleted Figures 2c and 2d in the original version since the key information of the VSc_2C cluster is given in the zoomed Figures 2e and 2f in the original version. Besides, we removed the bond angles in new Figures 2c and 2d for clarity, while the bond angles were given in Supplementary Information Figures 6 and 7 (new version).

Fig. R4 (added as new Supplementary Figure 8). (a, b) Two views of the $VSc_2C@D_{5h}(6)-C_{80}$. (a) Looking along the C_5 axis of cage. (b) Looking perpendicular to the C_5 axis of cage which is vertical in this view. (c, d) Two views of the $VSc_2C@D_{3h}(5)-C_{78}$. (c) Looking along the C_3 axis of cage. (d) Looking perpendicular to the C_3 axis of cage which is vertical in this view. The carbon atoms with labels show the location of the C_5/C_3 axis within $D_{5h}(6)-C_{80}/D_{3h}(5)-C_{78}$ cage.

Comment 5: A second aspect of the paper is the discussion of an “expanded Octet Rule” for endohedrals containing a central carbide ion. I did not find this topic to have much merit. Generally the term ‘expanded octet’ was used for situations where valence bond theory suggested the participation of d orbitals for elements in the second or lower in the periodic table. Since carbon only has s and p electrons available in the valence shell, the octet rule pertains to carbon period!

Answer: We thank the Reviewer for the critical comment with detailed explanation of the term “expanded Octet Rule”. We carefully made a literature survey and confirmed that the concept of an atom with an “expanded octet” usually refers to a hypervalent atom, whose structure may indicate the availability of low-lying d-orbitals [*J. Chem. Educ.* 76, 1013 (1999)]. Hence, using “expanded Octet Rule” in our present case is indeed inappropriate and may cause misleading problem, **although our initial meaning is to “expand” the concept of “Octet Rule” used typically for covalent compounds to metal carbido complexes exemplified by μ_3 -carbido clusterfullerenes (μ_3 -CCFs).**

Considering the claim of Reviewer 1, to avoid misunderstanding, we changed the term of “expanded Octet Rule” to “Supplemental Octet Rule” (which is now solidified with additional theoretical studies, see details of our answer to Comment 1 of Reviewer 2), i.e., **the classic Octet Rule commonly used for covalent compounds can now be applied for metal carbido complexes as demonstrated by our work and is thus regarded as “Supplemental”.** Besides, we changed the title to “*A stabilization rule for metal carbido cluster bearing μ_3 -carbido single-atom-ligand encapsulated in carbon cage*” to emphasize that the motivation of this work is to establish a general rule governing the stabilities of μ_3 -CCFs, which can not only interpret the stabilities of all μ_3 -CCFs reported so far but also would guide the exploration of novel μ_3 -CCFs or even other metal carbido complexes. In this way, we also clarify that the established “Supplemental Octet Rule” is applicable for the μ_3 -carbido single-atom-ligand coordinated with specific metals encapsulated within μ_3 -CCFs, thus the coincident misleading problem of “expanded Octet Rule” mentioned by this Reviewer can be excluded, since we focus on metal carbido cluster instead of covalent compounds containing elements in the second or lower row in the periodic table bearing d orbitals.

Comment 6: If one considered the central carbon as a carbide ion, then having any number of metal ions surrounding it presents no problems. The authors fail to make a convincing case that they can add anything of substance to our understanding of carbide-containing endohedrals in this paper.

Answer: We respectfully disagree with the Reviewer since we feel that the novelty and significance of our work might have been overlooked. Although for conventional metal carbido complexes the central carbon ion surrounded by 1 to 6 metal ions has been

reported as mentioned in the Introduction section, **the motivation of our work is to establish a general rule governing the stabilities of μ_3 -CCFs, which can not only interpret the stabilities of all reported μ_3 -CCFs reported so far but also would guide the exploration of novel μ_3 -CCFs or even other metal carbido complexes.** For this purpose, we first synthesized, isolated and characterized three novel V-based μ_3 -CCFs with unique structural feature such as V=C double bonds and high valence state of V (+4), which stimulates us to further carry out a systematic DFT study of all 16 reported μ_3 -CCFs, resulting in deduction of a general rule (“Supplemental Octet Rule”) governing the stabilities of μ_3 -CCFs for the first time. Hence, our focus is not the central carbide ion itself and the number of surrounding metal ions, instead we **focus on metal carbido cluster and manage to offer in-depth understanding of its unique structure and stability especially** the necessity of involving non-rare earth metal and formation of double bond between non-rare earth metal and the central carbon so as to satisfy the eight-electron configuration of the μ_3 -carbido ligand. To our knowledge, stabilities of organometallic complexes including metal carbido complexes have been commonly determined by Effective Atomic Number (EAN) rule (i.e., 18-electron rule) proposed in the 1920s, by studying μ_3 -CCFs as the simplest metal carbido complexes, our work demonstrates that the “Octet Rule” commonly used for covalent compounds can be also applied for metal carbido complexes exemplified by μ_3 -CCFs. **Our finding with the establishment of “Supplemental Octet Rule” undoubtedly extends our understanding of the “Octet Rule”, and is thus of high importance for μ_3 -CCFs and even metal carbido complexes.** We hope that the Reviewer understands its significance.

Reviewer #2 (Remarks to the Author):

The authors present some newly synthesised cage compounds. I am not from the field but these compounds are certainly interesting and challenging to synthesise. It is noted that the newly synthesised compound has an octet configuration on the central μ_3 carbon and this is used in an attempt to derive an expanded octet rule. Generally speaking, the compound synthesised is interesting. However, I don't believe that from the evidence presented one can really derive a general rule. The paper is generally well-presented but there is a bit too much jargon to reach a general audience.

We thank this Reviewer for her/his positive reviews with stimulating comments, which helped us to improve our manuscript.

Comment 1: If the authors really derived a general expanded octet rule, then they should explain its applications and its limits. One compound, present in several different cages, can hardly be used to derive a new rule. Can the authors explain unexplained trends in the literature or make new predictions using their rule?

Answer: We thank the Reviewer for the instructive suggestion. We agree with the Reviewer that more compounds are needed to deduce a new rule and it is important to discuss the applications and limits of this rule. In our original version, including three novel V-based μ_3 -CCFs, we just chose five representative μ_3 -CCFs based on three different types in terms of the non-rare earth metal (V, Ti, U), whose crystallographic structures are reported, to derive the general rule (see Supplementary Table 6, original version). Although three different types of metal carbido clusters are considered, it seems indeed inadequate.

In fact, except for these five representative μ_3 -CCFs, so far 16 μ_3 -CCFs in total have been reported, including other 11 Ti-based μ_3 -CCFs $\text{TiM}_2\text{C}@I_h(7)\text{-C}_{80}$ (M = Y, Nd, Gd, Tb, Dy, Er, Lu) whose crystallographic structures are unavailable (*Nat. Commun.* **5**, 3568–3576 (2014), *Angew. Chem. Int. Ed.* **54**, 13411–13415 (2015), *Inorg. Chim. Acta* **468**, 203–208 (2017)), $\text{TiDyYC}@I_h(7)\text{-C}_{80}$ (*Chem. Commun.* **54**, 10683–10686 (2018)), $\text{TiM}_2\text{C}@D_{5h}(6)\text{-C}_{80}$ (M = Sc, Dy) (*Angew. Chem. Int. Ed.* **54**, 13411–13415 (2015), *Chem. Eur. J.* **22**, 13098–13107 (2016)), $\text{TiSc}_2\text{C}@C_{78}$ (*Chem. Eur. J.* **22**, 13098–13107 (2016)). Considering the Reviewer's suggestion, we carried out additional theoretical calculations of the relative energies of other 11 reported Ti-based μ_3 -CCFs with different electronic configurations (see Table R5). According to these

new results, clearly we can see that, for all 11 μ_3 -CCFs, configuration **A** bearing a central carbon with an eight-electron configuration is the most stable structure for all cases. Therefore, combined with the original 5 μ_3 -CCFs, we confirm that the “Supplemental Octet Rule” is a general rule for μ_3 -CCFs.

Table R5 (added as Supplementary Table 9). The relative total energy (ΔE , eV) of isolated $\text{TiM}_2\text{C}@C_{2n}$ ($M = \text{Lu}, \text{Y}, \text{Nd}, \text{Gd}, \text{Tb}, \text{Dy}, \text{Er}, \text{Sc}$, $2n = 78, 80$) with different electronic configurations and M=C bonds of TiM_2C cluster.

	$\text{TiM}_2\text{C}@C_{2n}\text{-A}$ [8e; B.N.=4; V.S.=−4] ^a	$\text{TiM}_2\text{C}@C_{2n}\text{-B}$ [8e; B.N.=3; V.S.=−4] ^a	$\text{TiM}_2\text{C}@C_{2n}\text{-C}$ [7e; B.N.=4; V.S.=−3] ^a	$\text{TiM}_2\text{C}@C_{2n}\text{-D}$ [7e; B.N.=3; V.S.=−3] ^a
Electronic configuration				$\text{TiLu}_2\text{C}@I_h(7)\text{-C}_{80}$	0	1.94	0.09	2.37
$\text{TiY}_2\text{C}@I_h(7)\text{-C}_{80}$	0	0.39	0.48	0.83
$\text{TiNd}_2\text{C}@I_h(7)\text{-C}_{80}$	0	1.85	0.49	2.24
$\text{TiGd}_2\text{C}@I_h(7)\text{-C}_{80}$	0	2.28	0.63	2.71
$\text{TiTb}_2\text{C}@I_h(7)\text{-C}_{80}$	0	2.00	0.25	2.13
$\text{TiDy}_2\text{C}@I_h(7)\text{-C}_{80}$	0	2.37	0.54	2.56
$\text{TiEr}_2\text{C}@I_h(7)\text{-C}_{80}$	0	2.13	0.55	2.43
$\text{TiDyYC}@I_h(7)\text{-C}_{80}$	0	2.09	0.43	2.29
$\text{TiDy}_2\text{C}@D_{5h}(6)\text{-C}_{80}$	0	2.07	0.42	2.39
$\text{TiSc}_2\text{C}@D_{5h}(6)\text{-C}_{80}$	0	0.95	0.39	2.36
$\text{TiSc}_2\text{C}@D_{3h}(5)\text{-C}_{78}$	0	1.26	0.36	2.49

^a 8e/7e means the total of valence electron of $\text{C}^{4-}/\text{C}^{3-}$ anion; B.N. and V.S. represent bond number and valence state, respectively. Except for the 8e-rule, the structure is stable when $\text{B.N.} + \text{V.S.} = 0$ (electroneutral). Hence, configurations **B**, **C**, **D** are all less stable than configuration **A**.

Given that this general rule is confirmed on the basis of additional study on all reported μ_3 -CCFs, we clarify briefly its applications and limits as follow.

- 1) It can interpret the stabilities of all μ_3 -CCFs reported so far, especially the necessity of involving non-rare earth metal (V, Ti, U) and formation of double bond between non-rare earth metal and the central carbon ($M^{4+}=C^{4-}$) to satisfy the eight-electron configuration of the μ_3 -carbido ligand.
- 2) It would guide the exploration of novel μ_3 -CCFs or even other metal carbido complexes. As discussed in the Introduction section in details, metal carbido complexes bearing single-carbon-atom ligand such as the active site of nitrogenase (Fe_7MoS_9C) provide ideal models of adsorbed carbon atoms in heterogeneous catalysis, thus establishing a new rule governing the stabilities of metal carbido complexes is important for understanding its structure-performance correlation. As the simplest metal carbido complexes, μ_3 -CCFs are quite unique among all reported endohedral fullerenes because it offers a template to encapsulate non-rare earth metal especially transition metal into fullerene cage (see *Chem. Rev.* **2013**, *113*, 5989-6113). In this sense, this rule is expected to guide the exploration of novel μ_3 -CCFs, offering opportunity to encapsulate other 3d-block transition metals, thus advancing the development of endohedral fullerenes toward versatile applications.
- 3) The limitation of this rule is that it is established for μ_3 -CCFs in our work, while its applicability for other metal carbido complexes needs to be testified. We already mentioned this issue in the original version “*Noteworthy, this extended Octet Rule is also applicable for the conventional binuclear metal carbido complexes containing a carbido bridge such as $L_nM=C=ML_n$ and $L_nM\equiv C-M'L_n$, but it is not always valid for μ_3 -carbido ligand within the conventional trinuclear metal carbido complexes⁴⁻⁸. Furthermore, in the metal carbido complexes bearing μ_5 - and μ_6 -carbido ligands, this Rule is not applicable any more because the central carbon atom is coordinated with more than four metals and thus is a hypervalency carbon^{9,10}. For the well-known metal carbido complex Fe_7MoS_9C as the active site of nitrogenase, although a central C^{4-} anion also exists, it violates the extended Octet Rule because the central C^{4-} anion bonds with six iron atoms to form a Fe_6C core^{2,38}.*” (see Page 12, lines 8-17, original version)

To clarify this issue, we added two sentences in the paragraphs discussing the Supplemental Octet Rule “*So far, there are sixteen μ_3 -CCFs in total have been reported, we further calculated the relative energies of the other eleven μ_3 -CCFs with*

different configurations. We find that, similar to the cases of $MSc_2C@C_{2n}$ ($M = V, Ti, U, 2n = 78, 80$), the configuration bearing a central carbon with an eight-electron configuration is the most stable structure for all μ_3 -CCFs (Supplementary Table 9).” (see Page 12, lines 10-14, revised version); “this Supplemental Octet Rule may be used as a simple guide for design of novel μ_3 -CCFs, offering opportunity to encapsulate other non-rare earth metals into fullerene cage.” (see Page 13, lines 14-16, revised version). Besides, we added Table R3 as new Supplementary Table 9 in Supplementary Information.

Comment 2: The cage compounds are described with their specific notation $VSc_2C@I_h(7)-C_{80}$, etc. It would be good, if possible, to reduce this kind of jargon and explain in words what is the significance of the different compounds discussed. Why is it important that the compound is not of I_h symmetry?

Answer: We thank the Reviewer for the carefulness on checking specific notation. In fact, the specific notation of endohedral metallofullerene of $MXY@C_{2n}$ was introduced by Smalley et al. in 1991 to illustrate that the left species “MXY” is encapsulated within the C_{2n} fullerene cage written in right (see *J. Phys. Chem.* **95**, 7564-7568 (1994), *Chem. Rev.* **113**, 5989-6113 (2013)). This notation has been widely used in fullerene community for more than 30 years, hence we cannot change it or reduce the usage of this jargon.

Concerning the question why non- I_h - C_{80} -based μ_3 -CCFs are important, it is because among all 16 reported μ_3 -CCFs most members are based on I_h - C_{80} cage, which is the most stable cage and has the highest yield as predicted theoretically and confirmed experimentally for trimetallic μ_3 -carbido clusterfullerenes (μ_3 -CCFs) and trimetallic nitride clusterfullerenes bearing intramolecular six-electron-transfer from the encapsulated cluster to fullerene cage (see *Chem. Rev.* **113**, 5989-6113 (2013)). For this reason, so far only few I_h - C_{80} -based μ_3 -CCFs with relatively high yields have been crystallographically determined. This limits the systematic study of μ_3 -CCFs to derive a general rule. It is thus important to investigate whether non- I_h - C_{80} -based μ_3 -CCFs, which have much lower yields than I_h - C_{80} -based μ_3 -CCFs, can be isolated with considerable amount for crystal growth or not. Our study succeeded in isolating and determining crystallographically two non- I_h - C_{80} -based μ_3 -CCFs ($VSc_2C@D_{5h}(6)-C_{80}$

and $VSc_2C@D_{3h}(5)-C_{78}$) for the first time, therefore is very important for establishing a general rule governing the stabilities of μ_3 -CCFs.

Comment 3: The authors should explain the level of theory used. Is PBE with a plane-wave basis really the optimal choice? Shouldn't one at least use a dispersion correction to model the cage?

Answer: We thank the Reviewer for the professional comment on the level of theory. Actually, we did perform benchmark simulations (both geometric optimization and electronic structure calculations) at three different levels of theory, including the PBE functional, the HSE06 hybrid functional, and at the GGA level plus a dispersion correction (DFT-D3 method). As an example, Figure R5 plots the spin-resolved molecular levels and the spatial distributions of the frontier molecular orbitals of $VSc_2C@D_{5h}(6)-C_{80}$ obtained by using the DFT-D3 method and the HSE06 functional. We found that the optimized cages are not sensitive to the adopted functional and the bond length differences are less than 0.03 Å. Moreover, the arrangement of frontier molecular orbitals and their spatial distribution as well as the spin density are almost unchanged for different methods, although the energy level spaces are different. Therefore, in this work, we presented these calculated DFT results at the GGA-PBE level with a plane-wave basis.

Figure R5. (a) The spin-resolved molecular levels and the spatial distribution of the frontier molecular orbitals of $VSc_2C@D_{5h}(6)-C_{80}$ calculated by using the DFT-D3 (left) and hybrid HSE06 functional (right) method. (b) The corresponding spin density distributions.

Comment 4: The authors should explain how they really derive the electronic configuration of the metal ions.

Answer: We thank the Reviewer for the constructive comment. The molecular magnetic moments (1.02 and 0.98 μ_B) of $VSc_2C@D_{5h}(6)-C_{80}$ and $VSc_2C@D_{3h}(5)-C_{78}$ mainly originate from the metal ions, according to the calculated spin density distribution (see Fig. 3a). This is also verified by the predicted V atomic magnetic moment of 1.15 and 1.05 μ_B in $VSc_2C@D_{5h}(6)-C_{80}$ and $VSc_2C@D_{3h}(5)-C_{78}$, respectively. This observation implies that there is an unpaired electron, mainly contributed by the $3d_{z^2}$ orbitals (see Supplementary Fig. 11, revised version), for V atom in $VSc_2C@D_{5h}(6)-C_{80}$ and $VSc_2C@D_{3h}(5)-C_{78}$, leading to the doublet ground states. These results indicate that the electronic configuration of the encapsulated V ion is $3d^1$. Compared with the electronic configuration of free V atom ($3d^34s^2$), we can conclude that the V atom loses four valence electrons from the V atom, leading to the valence state of V^{4+} within both μ_3 -CCFs.

Comment 5: The authors say that the "molecules possess the spin-split ground states." It should be explained what is meant by that. Spin-orbit coupling? Or is it because broken symmetry DFT is used?

Answer: We thank the Reviewer for the carefulness on the term. We apologize for this confused statement. Actually, we mean that the two molecules possess the spin-polarized ground states, since the calculated total energy using the spin-restricted method is higher by about 0.35 eV than that of the spin-polarized scheme. To clarify this issue, we corrected this sentence to "*It is clear that two molecules possess the spin-polarized ground states.*" (see Page 9, lines 4-5 from the bottom, revised version)

Reviewer #3 (Remarks to the Author):

Herein, the authors propose a new rule that would govern the stabilities of metal carbide complexes. After having synthesized, purified, and characterized 3 new vanadium-based μ_3 CCFs, the authors are seeking and proposing an explanation for their unique electron configuration. In their TiSc₂C clusters, they discuss an expanded octet from the high valence state of a Ti⁴⁺ encapsulated metal cluster. The authors iteratively blend experiments and DFT calculations to support their rationale for an expanded octet rule for the μ_3 -carbido ligand in μ_3 -VSc₂C based CCFs.

One of the novel features of this paper is the peculiar bonding of the encapsulated metal carbide complex. Even the authors themselves address the novelty perspective with their statement on page 12: "In fact, the classic Octet Rule proposed by Lewis in 1916 has been commonly used for covalent compounds, but never been used for metal carbido complexes."

Indeed, one of the intriguing features of endohedral metallofullerenes is the opportunity given to us by nature for creating exotic bonding types engendered by the encapsulated cluster within the carbon fullerene cage housing. With a reputation for offering unique and uncommon bonding types stemming from endohedral clusters, such an expanded octet proposition is indeed credible. In short, the credibility of the author's proposal of an expanded Octet Rule for a μ_3 -carbido single-atom-ligand stabilized via the carbon cage is enhanced by such a unique molecular architecture as found in endohedral metallofullerenes.

Strengths:

The authors have synthesized, purified, and characterized 3 new species not known to exist, much less purified. The X-ray crystal structures were essential. Combined with DFT calculations, the authors have a scientifically sound basis for their proposed explanation of the expanded octet.

The results are valid and sound. Indeed, the amount of data and methodology from a variety of sources is very extensive. After reading carefully the manuscript and even the supplementary information, I do not find any glaring weak areas, and I have no suggested additional improvements. The iterative blend of DFT support and experimental data fortifies the credibility and soundness of the proposed expanded octet.

Conclusion:

The authors' contribution to Nature Chemistry has a high significance and novelty factor. I believe there to be interest to multiple fields of science. Indeed, the broad interest is address well on page 2 of the introduction. Moreover, there is an impressive amount of scientific support for their proposal of an expanded Octet Rule for these VSc₂C endohedral metallofullerenes. In totality, my opinion is in support of publication in Nature Communications.

We thank the Reviewer for the very positive comments with recognition of the novelty and significance of our work. With her/his strong support, we believe that our present work will not only bring an in-depth understanding of μ_3 -CCFs but also guide the exploration of novel μ_3 -CCFs or even other metal carbido complexes.

Reviewers' Comments:

Reviewer #1:

Remarks to the Author:

In this revision the authors have partially answered some of the issues raised about the crystallography. More information has been supplied, which I appreciate, but old problems remain and new problems now arise.

1. Overall, I remain unconvinced that the assignments of the V and Sc positions are correct, and one may never obtain a definitive result. The problem I identified earlier has not been addressed: It is possible and very likely that Sc and V may occupy the same site. How can one tell? The authors have not addressed this issue.

2. The authors have attempted to assign V and Sc positions on the basis of C-M bond distances, but as seen below for one example, there is a fairly gradual increase in these bond distances. I remain unconvinced that V and Sc can be differentiated in such highly disordered endohedrals.

VSc2C@D5h(6)-C80

1.812(11) V1c-C81

1.885(6) V1-C81

1.905(10) V1b-C81

1.920(9) Sc2a-C81

1.920(9) Sc2b-C81

1.923(5) V1a-C81

2.078(6) Sc2b-C81

2.162(9) Sc1c-C81

2.176(8) Sc2c-C81

2.186(5) Sc2-C81

2.226(6) Sc1a-C81

2.264(7) Sc1b-C81

3. The authors claim that their differentiation between Sc and V sites is substantiated by lower R factors for the assignment they favor, but the differences in R factors reported in Tables S5 and S6 are slight, while the R factors themselves are larger than one would like to report.

4. The data in table ST-3 indicates that the authors constrained their crystallographic refinement so that the occupancies of sets of three metal atoms were made equal. The better way to address the problem without bias would be to allow the metal site occupancies to refine freely. I would like to see the results of a free refinement model. The present version is quite biased.

In regard to the use of the well-established octet rule, I see nothing new here. The octet rule for carbon is taught in freshman chemistry and has provided insight into chemical bonding for many decades. The authors have simply backtracked on their terminology from their misleading "expanded

octet rule" to a "supplemental Octet Rule".

Overall, this is a routine endohedral fullerene paper that reports the preparation and purification of three new molecules. The analysis of the crystallographic data leaves much to be desired. The application of the octet rule for carbon lacks novelty or substance. I do not recommend publication of this article in Nature Communications. Publication in a more specialized venue might be possible after a more thorough and unbiased analysis of the crystallographic data.

Reviewer #2:

Remarks to the Author:

The authors have responded to all comments in detail. I think this is challenging, interesting and well-performed work.

I believe that this work warrants publication in a prominent format. But I am not from the field to say precisely how much impact and novelty this has.

Reviewer #3:

Remarks to the Author:

In response to the revised manuscript having been submitted, I am satisfied with the authors' responses to all 3 reviewers. Clearly, the authors have made an above and beyond effort to address all comments from all reviewers. The revised manuscript and supporting information is much better than the initial submission.

Recommendation: Proceed with publication

Point-by-point response to reviewer's comments:

Reviewer #1 (Remarks to the Author):

In this revision the authors have partially answered some of the issues raised about the crystallography. More information has been supplied, which I appreciate, but old problems remain and new problems now arise.

***Comment 1:** Overall, I remain unconvinced that the assignments of the V and Sc positions are correct, and one may never obtain a definitive result. The problem I identified earlier has not been addressed: It is possible and very likely that Sc and V may occupy the same site. How can one tell? The authors have not addressed this issue.*

Answer: We thank the Reviewer for recognizing our efforts and feel regretful that the Reviewer was not convinced by our detailed interpretation given in last round revision. We respectfully disagree with the Reviewer based on the reasons given in our rebuttal in last round revision along with the following reasons. To avoid repeated interpretation, we clarify this issue briefly as follow.

- 1) In our previous work (*Proc. Natl. Acad. Sci. U. S. A.* **119**, e2202563119 (2022), ref. 19), we already succeeded in determining the molecular structure of $VSc_2C@I_h(7)-C_{80}$ with reasonable assignments/differentiation of Sc and V sites by using the same method used in the present work. As explained in details in our rebuttal in last round revision, although it is indeed challenging to differentiate Sc and V crystallographically in non-fullerene molecules (without specific requirement on how they are combined together) because of their similar atomic numbers and similar scattering as the Reviewer said, fortunately differentiation of Sc and V sites within V-based μ_3 -carbido clusterfullerenes (μ_3 -CCFs) is feasible in our case **because of the requirements of specific metal-carbon bondings (one double bond plus two single bonds) and planar VSc_2C cluster**. Briefly, our method to differentiate Sc and V sites within V-based μ_3 -CCFs is based on two aspects: 1) the shorter bond must be $V=C$ double bond (due to V^{4+} configuration) and the longer bonds must be two Sc-C single bonds, thus the distinct discrepancy in bond lengths of $V=C$ double bonds and Sc-C single bonds facilitates the unambiguous assignments of Sc and V sites (see Fig. 2 and Fig. R1); 2) altering the V/Sc positions would result in increases of both R1 and wR2 values and less stable structures as confirmed by DFT calculations (see Supplementary Tables 5, 6). Although the

difference of R1 and wR2 values among three different configurations is relatively small due to the similarity on atomic number and scattering of V and Sc atom as the Reviewer claimed, such difference does exist and indicates that **the V/Sc positions are irreplaceable (Sc and V atoms CANNOT occupy the same site).**

Fig. R1. Major and minor orientations including the respective Sc/V-C bond lengths of VSc_2C cluster within $\text{VSc}_2\text{C}@I_h(7)\text{-C}_{80}$ (a), $\text{VSc}_2\text{C}@D_{5h}(6)\text{-C}_{80}$ (b) and $\text{VSc}_2\text{C}@D_{3h}(5)\text{-C}_{78}$ (c). Note that, although there may be a slight overlap between Sc and V sites in the minor orientations B, C of $\text{VSc}_2\text{C}@D_{5h}(6)\text{-C}_{80}$ in terms of the short Sc2a-C and Sc2b-C bond lengths (see our answer to Comment 2 below for details), the major and most minor orientations all meet the requirement of specific metal-carbon bondings (one double bond [V=C] plus two single bonds [Sc-C]).

2) **The method to differentiate two adjacent metal atoms we used in our present work has been widely used in literatures for other endohedral metallofullerenes.** For example, another Ti-based $\mu_3\text{-CCF TiSc}_2\text{C}@I_h(7)\text{-C}_{80}$, in which Sc and Ti have even more close atomic numbers and scattering than those between Sc and V, was unambiguously determined by single-crystal X-ray diffraction with the distinct difference between Ti=C double bond and Sc-C single bond by Popov et al. (*Chem. Eur. J.* **22**, 13098-13107 (2016), ref. 13). Even for the trimetallic nitride clusterfullerenes based on encapsulation of $\text{VSc}_2\text{N}/\text{V}_2\text{ScN}$ cluster bearing three Sc(V)-N single bonds within $\text{VSc}_2\text{N}@I_h(7)\text{-C}_{80}$, $\text{VSc}_2\text{N}@D_{5h}(6)\text{-C}_{80}$, and $\text{V}_2\text{ScN}@I_h(7)\text{-C}_{80}$, Sc/V atoms have been successfully differentiated based on even smaller difference between V-N bond lengths and Sc-N bond lengths (*J. Am. Chem. Soc.* **138**, 207-214 (2016), *Angew. Chem. Int. Ed.* **57**, 10273-10277 (2018), refs. 22, 23).

3) Following the Reviewer's suggestion "allow the metal site occupancies to refine freely" in Comment 4, we refined the crystallographic structures of $VSc_2C@D_{5h}(6)-C_{80}$ and $VSc_2C@D_{3h}(5)-C_{78}$ with free refinement models of encapsulated metals. As analyzed in details in our answer to Comment 4, according to the results of free refinement model, the bond lengths of the corresponding disordered metal sites to the central carbon do not change significantly, and the distinct difference between V=C bond lengths and Sc-C bond lengths are still observed. However, the metals (Sc1/Sc2/V1) with the major occupancies are not capable of forming a planar and rational triangular configuration of VSc_2C cluster within $VSc_2C@D_{3h}(5)-C_{78}$. Hence, **the results of a free refinement model are unreasonable**. This is also the reason why **the constraints of site occupancies are very common in crystallographic refinements of endohedral metallofullerenes** including other μ_3 -CCFs ($TiLu_2C@I_h(7)-C_{80}$: *Nat. Commun.* **5**, 3568-3576 (2014); $TiTb_2C@I_h(7)-C_{80}$: *Inorg. Chim. Acta* **468**, 203-208 (2017); $USc_2C@I_h(7)-C_{80}$: *Chem. Commun.* **56**, 3867-3870 (2020)) and other types of endohedral metallofullerenes (see e.g., $Sc_3N@C_{2v}(7854)-C_{70}$: *J. Am. Chem. Soc.* **143**, 612-616 (2021); $U_2N@I_h(7)-C_{80}$: *Chem. Sci.* **12**, 282-292 (2021); $Sc_3N@C_s(39663)-C_{82}$: *Chem. Commun.* **57**, 4150-4153 (2021); $Ho_2C_2@C_2(61)-C_{92}$: *Inorg. Chem.* **61**, 605-612 (2022); $Tb_2@I_h(7)-C_{80}(CF_3)$: *J. Am. Chem. Soc.* **143**, 18139-18149 (2021); $Sc_2S@C_s(10528)-C_{72}$: *J. Am. Chem. Soc.* **134**, 7851-7860 (2012); $Sc_2O@C_{2v}(5)-C_{80}$: *Inorg. Chem.* **54**, 9845-9852 (2015); $Sc_2O@T_d(19151)-C_{76}$: *Chem. Eur. J.* **21**, 11110-11117 (2015); $Dy_2O@C_s(6)-C_{82}$: *Adv. Sci.* **6**, 1901352 (2019); $Ho_2O@C_2(13333)-C_{74}$: *Inorg. Chem.* **58**, 4774-4781 (2019); $Ho_2O@D_{2d}(51591)-C_{84}$: *Inorg. Chem.* **58**, 10905-10911 (2019)).

Based on these reasons, we believe that our structure solutions including especially the assignments of Sc and V sites within μ_3 -CCFs in our present work are reliable, and Reviewers 2 and 3 did not argue this issue at all. **The method to differentiate two adjacent metal atoms we used in our present work has been commonly accepted in fullerene community**. We do hope that the Reviewer can understand it.

Comment 2: The authors have attempted to assign V and Sc positions on the basis of C-M bond distances, but as seen below for one example, there is a fairly gradual increase in these bond distances. I remain unconvinced that V and Sc can be differentiated in such highly disordered endohedrals.

VSc₂C@D_{5h}(6)-C₈₀

1.812(11) V1c-C81

1.885(6) V1-C81

1.905(10) V1b-C81

1.920(9) Sc2a-C81

1.920(9) Sc2b-C81

1.923(5) V1a-C81

2.078(6) Sc2b-C81

2.162(9) Sc1c-C81

2.176(8) Sc2c-C81

2.186(5) Sc2-C81

2.226(6) Sc1a-C81

2.264(7) Sc1b-C81

Answer: We thank the Reviewer for the carefulness on checking the metal-C bond distances. In our last round revised version, we already listed all V-C and Sc-C bond lengths (see Supplementary Table 4). As depicted in Fig. R1, there are 22 disordered Sc sites and 22 corresponding Sc-C bonds in three V-based μ_3 -CCFs. It can be seen that most of Sc-C bond lengths exceed 1.99 Å, which are distinctly larger than V=C bond lengths (ranging from 1.8-1.92 Å). Among them, only Sc2a-C bond length (1.956(6) Å) and Sc2b-C bond length (1.920(9) Å) within *VSc₂C@D_{5h}(6)-C₈₀* are much smaller than 1.99 Å as exceptional cases. Even though these two values are smaller than other Sc-C bond lengths within *VSc₂C@D_{5h}(6)-C₈₀*, they are still larger than the corresponding V-C bond lengths (see Fig. R1). Besides, this phenomenon can be also seen in other endohedral metallofullerenes reported in literatures. For example, for the recently reported U-based μ_3 -CCF *USc₂C@I_h(7)-C₈₀* (*Chem. Commun.* **56**, 3867-3870 (2020)), there are four orientations of *USc₂C* cluster in *USc₂C@I_h(7)-C₈₀*, in which bond lengths of Sc-C single bond are in the range of 1.90-2.20 Å, with three Sc-C bond lengths (Sc1a-C, Sc2b-C, Sc2c-C) smaller than 1.97 Å (Fig. R2).

Fig. R2. Major and minor orientations including the respective Sc/U-C bond lengths of USc_2C cluster within $USc_2C@I_h(7)-C_{80}$. The data is extracted from *Chem. Commun.* **56**, 3867-3870 (2020).

In fact, among 22 Sc-C bonds corresponding to the respective disordered Sc sites in three V-based μ_3 -CCFs, only one Sc-C bond length is smaller than 1.923 Å, which is the maximum V=C bond length within V-based μ_3 -CCFs, while the other 21 Sc-C bond lengths are much larger than 1.923 Å (19 of them exceed 2 Å). Therefore, one or two exceptional Sc-C bond lengths are in the acceptable level and do not affect our statement on the general discrepancy in bond lengths between Sc-C single bond and V=C double bond.

Comment 3: *The authors claim that their differentiation between Sc and V sites is substantiated by lower R factors for the assignment they favor, but the differences in R factors reported in Tables S5 and S6 are slight, while the R factors themselves are larger than one would like to report.*

Answer: We thank the Reviewer for the carefulness on checking the R factors. In Supplementary Tables 5 and 6, by comparing three possible conformations of $VSc_2C@D_{5h}(6)-C_{80}$ and $VSc_2C@D_{3h}(5)-C_{78}$ with altered V/Sc positions, we wanted to clarify the most stable conformations. In fact, the R1 and wR2 values are comparable to those reported for other endohedral metallofullerenes in literatures as commonly accepted in fullerene community (see Table R1). Although the difference of R1 and wR2 values among three different configurations is relatively small due to the similarity on atomic number and scattering of V and Sc atom, such difference does exist and indicates that **the V/Sc positions are irreplaceable (Sc and V atoms CANNOT occupy the same site)**. In addition, our conclusion that altering the V/Sc positions would result in increases of both R1 and wR2 values and less stable structures are confirmed by DFT calculations, solidifying the reliability of our method to differentiate Sc and V sites within V-based μ_3 -CCFs.

Table R1. R1 and wR2 values of representative reported endohedral metallofullerenes.

Compound	R1	wR2	Reference
U ₂ @I _h (7)-C ₈₀	0.1169	0.3706	J. Am. Chem. Soc. 140 , 3907–3915 (2018)
Sc ₂ O@T _d (19151)-C ₇₆	0.1218	0.3433	Chem. Eur. J. 21 , 11110–11117 (2015)
Dy ₂ O@C _{2v} (9)-C ₈₂	0.1234	0.3527	Adv. Sci. 6 , 1901352 (2019)
Dy ₂ C ₂ @C _s (32)-C ₈₈	0.1286	0.3999	Inorg. Chem. Front. 9 , 5805–5819 (2022)
Dy ₂ O@C _{3v} (8)-C ₈₂	0.1323	0.4232	Adv. Sci. 6 , 1901352 (2019)
Eu@C _{2v} (19138)-C ₇₆	0.1334	0.3425	Angew. Chem. Int. Ed. 59 , 5259–5262 (2020)
Lu ₂ C ₂ @C ₂ (41)-C ₉₀	0.1391	0.3626	Carbon 164 , 157–163 (2020)
Er ₂ C ₂ @C _s (32)-C ₈₈	0.1427	0.3448	Inorg. Chem. 59 , 1940–1946 (2020)
Dy ₂ O@C _s (6)-C ₈₂	0.1479	0.4042	Adv. Sci. 6 , 1901352 (2019)
Dy@D ₂ (21)-C ₈₄	0.1727	0.3734	Inorg. Chem. Front. , 10 , 4139–4146 (2023)
Er ₂ C ₂ @C _s (6)-C ₈₂	0.1752	0.4088	Inorg. Chem. 59 , 1940–1946 (2020)
VSc ₂ C@I _h (7)-C ₈₀	0.0854	0.2313	Proc. Natl. Acad. Sci. U. S. A. 119 , e2202563119 (2022)
VSc ₂ C@D _{5h} (6)-C ₈₀	0.0865	0.2634	This work
VSc ₂ C@D _{3h} (5)-C ₇₈	0.1205	0.3524	This work

Comment 4: The data in table ST-3 indicates that the authors constrained their crystallographic refinement so that the occupancies of sets of three metal atoms were made equal. The better way to address the problem without bias would be to allow the metal site occupancies to refine freely. I would like to see the results of a free refinement model. The present version is quite biased.

Answer: Following the Reviewer's suggestion, we refined the crystallographic structures of VSc₂C@D_{3h}(5)-C₇₈ and VSc₂C@D_{5h}(6)-C₈₀ with free refinement models of encapsulated metals. As shown in Figs. R3 and R4, for the structures obtained by free refinement models, the bond lengths of the corresponding disordered metal sites to the central carbon do not change significantly, and the distinct difference between V=C bond lengths and Sc-C bond lengths are still observed. However, the metals (Sc1/Sc2/V1) with the major occupancies are not capable of forming a planar and rational triangular configuration of VSc₂C cluster within VSc₂C@D_{3h}(5)-C₇₈ (see Fig. R3d). In fact, the reported Ti-based μ₃-CCF TiSc₂C@I_h(7)-C₈₀ and U-based μ₃-CCF USc₂C@I_h(7)-C₈₀ both adopt the planar configurations of the encapsulated MSc₂C clusters (*Chem. Eur. J.* **22**, 13098-13107 (2016); *Chem. Commun.* **56**, 3867-3870

(2020)), thus the planar triangular configurations of VSc₂C clusters within V-based μ_3 -CCFs are expected, as confirmed by DFT calculations. Hence, **the results of a free refinement model are unreasonable in this case.** These results suggest that the disorder of each metal atom is derived from the rotation of the entire planar VSc₂C cluster, rather than the vibration of individual metal. This phenomenon is also observed in Ti-based μ_3 -CCFs TiLu₂C@I_h(7)-C₈₀, TiTb₂C@I_h(7)-C₈₀ and U-based μ_3 -CCF USc₂C@I_h(7)-C₈₀ (*Nat. Commun.* **5**, 3568-3576 (2014); *Inorg. Chim. Acta* **468**, 203-208 (2017); *Chem. Commun.* **56**, 3867-3870 (2020)). For the case of VSc₂C@D_{5h}(6)-C₈₀, the structure obtained by free refinement models looks quite similar to that obtained by constrained refinement model with slight difference on bond lengths and bond angles (see Fig. R4b,d). For consistency, we prefer to keep the previous crystallographic structures of VSc₂C@D_{3h}(5)-C₇₈ and VSc₂C@D_{5h}(6)-C₈₀ obtained by constrained refinement model. Since Reviewers 2 and 3 did not argue this issue, we hope that this Reviewer can understand it.

Fig. R3. (a) Positions of the disordered metal sites with site occupancy (number aside the atom) and bond lengths between metal sites and the central carbon atom in VSc₂C@D_{3h}(5)-C₇₈ obtained by constrained refinement model. (b) The positions of the major metal sites with the largest

occupancies within $VSc_2C@D_{3h}(5)-C_{78}$ obtained by constrained refinement model. (c) Positions of the disordered metal sites with site occupancy (number aside the atom) and bond lengths between metal sites and the central carbon atom in $VSc_2C@D_{3h}(5)-C_{78}$ obtained by free refinement model. (d) The positions of the major metal sites with the largest occupancies by $VSc_2C@D_{3h}(5)-C_{78}$ obtained by free refinement model.

Fig. R4. (a) Positions of the disordered metal sites with site occupancy (number aside the atom) and bond lengths between metal sites and the central carbon atom in $VSc_2C@D_{5h}(6)-C_{80}$ obtained by constrained refinement model. (b) The positions of the major metal sites with the largest occupancies within $VSc_2C@D_{5h}(6)-C_{80}$ obtained by constrained refinement model. (c) Positions of the disordered metal sites with site occupancy (number aside the atom) and bond lengths between metal sites and the central carbon atom in $VSc_2C@D_{5h}(6)-C_{80}$ obtained by free refinement model. (d) The positions of the major metal sites with the largest occupancies within $VSc_2C@D_{5h}(6)-C_{80}$ obtained by free refinement model.

Indeed, **the constraints of site occupancies are very common in crystallographic refinements of endohedral metallofullerenes** including other μ_3 -CCFs ($TiLu_2C@I_h(7)-C_{80}$: *Nat. Commun.* **5**, 3568-3576 (2014); $TiTb_2C@I_h(7)-C_{80}$: *Inorg. Chim. Acta* **468**, 203-208 (2017); $USc_2C@I_h(7)-C_{80}$: *Chem. Commun.* **56**, 3867-3870 (2020)) and other types of endohedral metallofullerenes (see e.g., $Sc_3N@C_{2v}(7854)-$

C₇₀: *J. Am. Chem. Soc.* **143**, 612-616 (2021); U₂N@I_h(7)-C₈₀: *Chem. Sci.* **12**, 282-292 (2021); Sc₃N@C_s(39663)-C₈₂: *Chem. Commun.* **57**, 4150-4153 (2021); Ho₂C₂@C₂(61)-C₉₂: *Inorg. Chem.* **61**, 605-612 (2022); Tb₂@I_h(7)-C₈₀(CF₃): *J. Am. Chem. Soc.* **143**, 18139-18149 (2021); Sc₂S@C_s(10528)-C₇₂: *J. Am. Chem. Soc.* **134**, 7851-7860 (2012); Sc₂O@C_{2v}(5)-C₈₀: *Inorg. Chem.* **54**, 9845-9852 (2015); Sc₂O@T_d(19151)-C₇₆: *Chem. Eur. J.* **21**, 11110-11117 (2015); Dy₂O@C_s(6)-C₈₂: *Adv. Sci.* **6**, 1901352 (2019); Ho₂O@C₂(13333)-C₇₄: *Inorg. Chem.* **58**, 4774-4781 (2019); Ho₂O@D_{2d}(51591)-C₈₄: *Inorg. Chem.* **58**, 10905-10911 (2019).

In regard to the use of the well-established octet rule, I see nothing new here. The octet rule for carbon is taught in freshman chemistry and has provided insight into chemical bonding for many decades. The authors have simply backtracked on their terminology from their misleading “expanded octet rule” to a “supplemental Octet Rule”.

Overall, this is a routine endohedral fullerene paper that reports the preparation and purification of three new molecules. The analysis of the crystallographic data leaves much to be desired. The application of the octet rule for carbon lacks novelty or substance. I do not recommend publication of this article in Nature Communications. Publication in a more specialized venue might be possible after a more thorough and unbiased analysis of the crystallographic data.

Answer: The Reviewer simply overlooked our efforts made in our last round revision (which are however well recognized by Reviewer 3 saying “*Clearly, the authors have made an above and beyond effort to address all comments from all reviewers.*”), hence we respectfully disagree with her/him. As emphasized repeatedly, the novelty of our present work is that **the classic Octet Rule commonly used for covalent compounds is applied for metal carbido complexes for the first time, which can not only interpret the stabilities of all reported μ_3 -CCFs reported so far but also would guide the exploration of novel μ_3 -CCFs or even other metal carbido complexes.** This significance has been well recognized by Reviewer 3 in the first round (“*One of the novel features of this paper is the peculiar bonding of the encapsulated metal carbide complex. Even the authors themselves address the novelty perspective with their statement on page 12...*”; “*...such an expanded octet proposition is indeed credible.*”) and by Reviewer 2 in the second round (“*I think this is challenging, interesting and well-performed work.*”). We do hope that Reviewer 1 can understand it.

Reviewer #2 (Remarks to the Author):

The authors have responded to all comments in detail. I think this is challenging, interesting and well-performed work.

I believe that this work warrants publication in a prominent format. But I am not from the field to say precisely how much impact and novelty this has.

We thank this Reviewer for her/his very positive comment with recognition of the novelty and significance of our work.

Reviewer #3 (Remarks to the Author):

In response to the revised manuscript having been submitted, I am satisfied with the authors' responses to all 3 reviewers. Clearly, the authors have made an above and beyond effort to address all comments from all reviewers. The revised manuscript and supporting information is much better than the initial submission.

Recommendation: Proceed with publication

We thank this Reviewer for her/his very positive comment with recognition of the novelty and significance of our work.

Reviewers' Comments:

Reviewer #3:

Remarks to the Author:

[Note from the Editor: Reviewer #3 was asked to assess the response given to reviewer #1 who was not able to look over the revision again.]

The authors have addressed my concerns. Moreover, in my opinion, the authors have address ALL of the reviewers concerns. In summary, I am satisfied and believer the work should be accepted and published in Nature Communications.